# New recognition specificity in a plant immune receptor by molecular engineering of its integrated domain

Stella Cesari [1✉], Yuxuan Xi [1], Nathalie Declerck[1], Véronique Chalvon[1], Léa Mammri [2], Martine Pugnière [3], Corinne Henriquet[3], Karine de Guillen [2], Vincent Chochois[4], André Padilla [2] & Thomas Kroj [1✉]

Plant nucleotide-binding and leucine-rich repeat domain proteins (NLRs) are immune sensors that recognize pathogen effectors. Here, we show that molecular engineering of the integrated decoy domain (ID) of an NLR can extend its recognition spectrum to a new effector. We relied for this on detailed knowledge on the recognition of the *Magnaporthe oryzae* effectors AVR-PikD, AVR-Pia, and AVR1-CO39 by, respectively, the rice NLRs Pikp-1 and RGA5. Both receptors detect their effectors through physical binding to their HMA (Heavy Metal-Associated) IDs. By introducing into RGA5_HMA the AVR-PikD binding residues of Pikp-1_HMA, we create a high-affinity binding surface for this effector. RGA5 variants carrying this engineered binding surface perceive the new ligand, AVR-PikD, and still recognize AVR-Pia and AVR1-CO39 in the model plant *N. benthamiana*. However, they do not confer extended disease resistance specificity against *M. oryzae* in transgenic rice plants. Altogether, our study provides a proof of concept for the design of new effector recognition specificities in NLRs through molecular engineering of IDs.

[1] PHIM Plant Health Institute, Univ. Montpellier, INRAE, CIRAD, Institut Agro, IRD, Montpellier, France. [2] CBS, Univ. Montpellier, CNRS, INSERM, Montpellier, France. [3] IRCM, Institut de Recherche en Cancérologie de Montpellier, INSERM U1194, Université de Montpellier, Institut Régional du Cancer de Montpellier, Montpellier F-34298, France. [4] Qualisud, Univ. Montpellier, Avignon Université, CIRAD, Institut Agro, Université de La Réunion, Montpellier, France. ✉email: stella.cesari@inrae.fr; thomas.kroj@inrae.fr

Nucleotide-binding (NB) and leucine-rich repeat domain (NLR) receptors are key elements of plant immunity. They detect the activity or the presence of specific pathogen-derived effector proteins that are secreted and trans-located inside host cells and activate defense responses and immunity[1]. Since they confer resistance to many crop diseases, which represent a major threat to agriculture, NLR-coding genes are widely used in crop breeding programs. Extending the ability of NLRs to recognize a broader range of pathogens is a challenge and represents a major goal for improved disease resistance in crops.

NLR receptors have a conserved architecture comprising a central NB domain, a C-terminal leucine-rich repeat (LRR) domain, and a variable N-terminal signaling domain that is most of the coiled-coil (CC) or the Toll-Interleukin1/Receptor (TIR) type[2]. NLRs recognize specific pathogen effectors through different molecular mechanisms: some interact physically with the recognized effector, while others perceive specific changes induced by an effector on a plant target protein or on a decoy protein that mimics the genuine effector target and thus serves as an effector trap[3,4]. Many plant NLRs harbor one or multiple non-canonical domains integrated into their structure[5–10]. Some of these integrated domains (ID) were shown to be involved in the specific recognition of effectors and thought to be decoy domains derived from proteins targeted by these effectors[11–16]. NLR-IDs often cluster genetically and function in combination with a second NLR that acts as a signaling executer for the sensor NLR-ID.

Due to the complexity of NLR function, very few studies have explored the potential of engineering NLR-mediated resistance in plants. Changes in effector recognition specificity between different alleles of the flax NLRs L or P were achieved through domain or residue swaps in the LRR domain[17–20]. Targeted point mutations or random mutational screens succeeded in extending NLR recognition specificities or increasing their activation properties to create sensitized NLRs[21–23]. An alternative approach is the engineering of decoy proteins. This was successful in the case of PBS1, which activates the NLR RPS5 upon its cleavage by a bacterial protease effector. The cleavage site of PBS1 was replaced by cleavage sites for other protease effectors from bacteria and viruses resulting in RPS5-dependent resistance to these pathogens[24,25]. Finally, a promising strategy consists of engineering IDs either to extend the recognition spectrum of NLRs or to create new specificities. Current examples of such engineering mainly focus on improving the ability of a particular NLR to recognize different alleles of a specific effector[26].

In rice, the sensor/executer NLR pair RGA5/RGA4 confers resistance to Magnaporthe oryzae isolates carrying the effector genes AVR1-CO39 or AVR-Pia, while the Pikp-1 and Pikp-2 NLR pair recognize isolates expressing AVR-PikD[11,27,28]. AVR1-CO39, AVR-Pia, and AVR-PikD are sequence-unrelated but possess highly similar β-sandwich structures characteristic of the Magnaporthe AVRs and ToxB-like (MAX) effector family in plant pathogenic Ascomycete fungi[12,29]. Both RGA5 and Pikp-1 sesnor NLRs contain a heavy metal-associated (HMA) ID that is crucial for specific effector recognition through direct binding[11,12]. These HMA domains share 54% sequence identity and are located at the C-terminus in RGA5 or between the CC and the NB domains in Pikp-1. Structure–function analyses provided detailed insight into RGA5_HMA/AVR-Pia, RGA5_HMA/AVR1-CO39, and Pikp-1_HMA/AVR-PikD binding and established a causal link between these interactions and effector recognition specificities[12,30,31].

Remarkably, AVR1-CO39 and AVR-PikD bind different surfaces of the HMA domains[12,31,32], suggesting high plasticity in ID-effector interactions[33]. AVR1-CO39 mainly interacts with the α1 helix and β2 strand of RGA5_HMA whereas AVR-PikD recognition by Pikp-1_HMA involves residue side-chains mainly located in β2, β3, β4, and the terminal K262 residue. Although the 3D structure of the AVR-Pia/RGA5_HMA complex has not been determined yet, gel filtration analyses and 3D modeling suggest that AVR-Pia binds RGA5_HMA through the same interface as AVR1-CO39[31] (Fig. 1). A crystal structure of the AVR-Pia/Pikp-1_HMA complex, showing that AVR-Pia interacts with Pikp-1_HMA through its α1 helix and β2 strand, further supports this hypothesis[33].

Recent studies have highlighted the potential of ID engineering by altering the recognition specificity of Pik-1 through structure-guided modifications of precise residues within the HMA domain[26]. This broadened the recognition specificity of the Pik-1 NLR, enabling the perception of multiple AVR-Pik alleles.

We hypothesized that by combining the AVR1-CO39 and AVR-PikD binding interfaces in the HMA domain of RGA5, we could generate an RGA5 variant able to bind and recognize both effectors as well as AVR-Pia. We show that the introduction of the AVR-PikD binding interface in RGA5_HMA expands the effector binding capacity of RGA5 in vivo and in vitro. The engineered RGA5 NLR functions in RGA4 repression and recognizes both AVR-Pia and AVR-PikD in the Nicotiana benthamiana heterologous system. However, it does not confer rice resistance to M. oryzae isolates expressing AVR-PikD although it still provides resistance to isolates carrying AVR-Pia or AVR1-CO39.

Our study, therefore, provides a proof of concept that structure-guided engineering is effective to create novel effector-binding interfaces and new recognition specificity for NLR-IDs. However, we also highlight and discuss important constraints and current limitations of the NLR-ID engineering strategy.

## Results

**Structure-guided engineering of the HMA domain of RGA5.** Based on the structure of the Pikp-1_HMA/AVR-PikD complex[12,32], we designed mutations in the HMA domain of RGA5 to enable AVR-PikD binding and recognition. The amino acid sequences of the HMA domains of RGA5 and Pikp-1 were aligned (Fig. 1a) and the residues of Pikp-1_HMA responsible for AVR-PikD binding were swapped by site-directed mutagenesis into RGA5_HMA generating three RGA5_HMA variants. The m1 mutant harbors nine substituted residues in the β-strands 2 and 3 (E1029A, I1030L, T1031V, E1033D, D1034L, K1035R, R1037K, L1038I, and V1039E). The m2 mutant carries three amino acid substitutions in the β4 strand (M1065Q, E1067S, and L1068Q) that were defined based on the first published Pikp-1_HMA/AVR-PikD structure[12]. The m1m2 mutant combines the m1 and m2 swaps and thereby harbors almost entirely the AVR-PikD-binding surface of Pikp-1.

The effector-binding surfaces of RGA5_HMA and Pikp-1_HMA are located on opposite sides of the β-sheet forming the conserved core of HMA domains and overlapping only slightly on β-strand 2 (Fig. 1a, b). Therefore, the m1, m2, and m1m2 mutants remain mostly unchanged for the residues that directly bind AVR1-CO39 (Fig. 1a). The structure of the RGA5_HMA/AVR-Pia complex has not been determined yet. However, functional data and the structure of a complex formed between AVR-Pia and Pikp1_HMA suggest a strong overlap between the AVR-Pia and the AVR1-CO39 binding surfaces in RGA5_HMA[31,33]. Structural modeling of the RGA5_HMA/AVR-Pia complex supports this hypothesis (Fig. 1c, Supplementary Table 1). Therefore, the m1, m2, and m1m2 mutations were expected not to interfere with the binding of AVR-Pia to RGA5_HMA.

To address more precisely the potential impact of the m1 and m2 mutations on binding to AVR-Pia, AVR1-CO39, and AVR-PikD, we modeled the structure of the complexes formed between

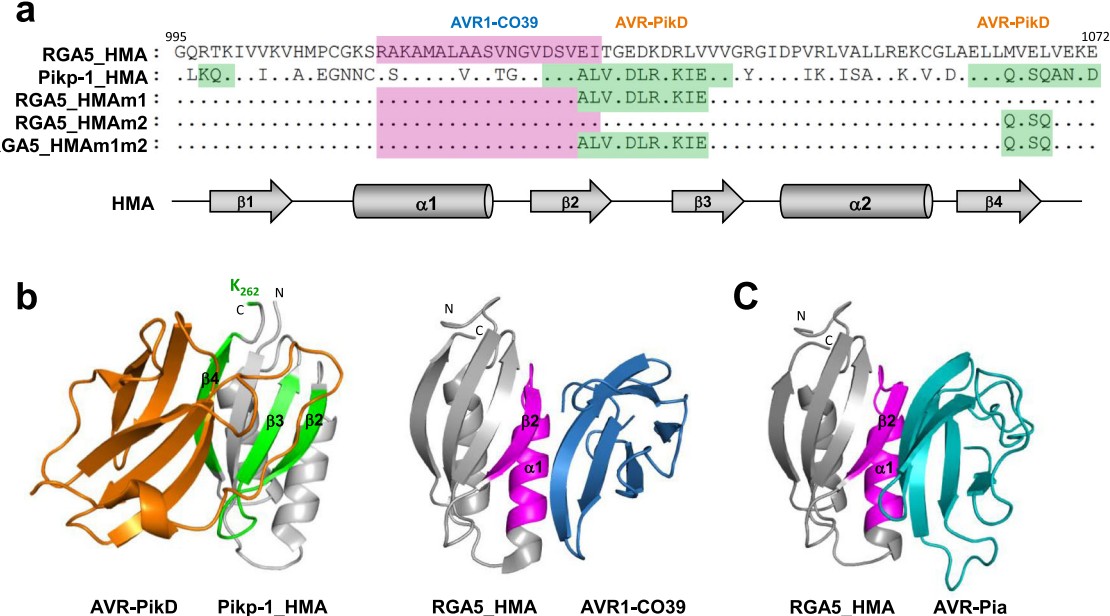

**Fig. 1 Structure-guided engineering of the HMA domain of RGA5 to introduce the AVR-PikD binding surface. a** Sequence alignment of the RGA5 and Pikp-1 HMA domains. The residues constituting the AVR1-CO39 and AVR-PikD binding interfaces are highlighted in pink and green, respectively. Residues in strands β2/β3 and β4 targeted respectively by the m1 and m2 mutations are highlighted in green in the different RGA5_HMA variants. Secondary structure elements of the HMA domain are shown below in gray. **b** Crystal structure of the AVR-PikD/Pikp-1_HMA (left) and AVR1-CO39/RGA5_HMA (right) complexes (PDB_6G10 and PDB_5ZNG, respectively). AVR-PikD (orange) binds the HMA domain of Pikp-1 through residues in β2/β3, β4, and K262 (green) and AVR1-CO39 (blue) interacts with the HMA domain of RGA5 through its α1/β2 surface (pink). **c** 3D model of RGA5_HMA in complex with AVR-Pia (light blue) based on the AVR-Pia/Pikp-1_HMA template structure (PDB_6Q76).

RGA5_HMAm1m2 and all three effectors and calculated corresponding binding energies (Supplementary Fig. 1, Supplementary Table 1). This provided similar binding interfaces and energies for the complexes of AVR-Pia and AVR1-CO39 with RGA5_HMA or RGA5_HMAm1m2. Binding parameters of AVR-PikD/RGA5_HMAm1m2 were similar to those in the AVR-PikD/Pikp_HMA complex in both binding surface and binding energy (Supplementary Fig. 1, Supplementary Table 1). Detailed in silico analysis, therefore, supports the hypothesis that the RGA5_HMA mutants retain the ability to bind AVR1-CO39 and AVR-Pia and that at least RGA5_HMAm1m2 has a high affinity for AVR-PikD.

**Engineered HMA domains of RGA5 bind AVR-PikD in yeast two-hybrid assays.** Using yeast-two-hybrid assays, we found that AVR-PikD interacts with the m1 and m1m2 mutants of both the isolated RGA5_HMA domain (residues 991–1072) and a longer C-terminal fragment of RGA5 (residues 883–1116) (Fig. 2). This RGA5_C-ter fragment includes the sequence downstream of the HMA and part of the linker connecting the LRR and HMA domains and has in vitro the same AVR-Pia and AVR1-CO39-binding characteristics as the isolated RGA5_HMA domain[31]. These interactions of AVR-PikD with the m1 and m1m2 mutant constructs are as strong as the one observed with Pikp-1_HMA as shown by yeast growth on stringent selective media supplemented with 10 mM of 3-amino-1,2,4-triazole (3AT). AVR-PikD does not bind RGA5_HMA and, as already described, binds very weakly RGA5_C-ter[11]. Both RGA5_HMA and RGA5_C-ter m2 mutants also fail to interact with AVR-PikD indicating that changing the corresponding residues in the RGA5_HMA β strand 4 are not sufficient for engineering strong binding. As previously reported, RGA5_C-ter binds strongly to AVR1-CO39 and AVR-Pia[11] but, unexpectedly, RGA5_HMA does not. The three RGA5_C-ter mutant variants interact with AVR-Pia almost at the same level as wild type RGA5_C-ter, but their interaction with AVR1-CO39 is

reduced. Indeed, the m1 and m2 mutations slightly decrease binding to AVR1-CO39, while the m1m2 strongly weakens this interaction but does not abolish it (Fig. 2, Supplementary Fig. 2). Overall, interactions observed for AVR-PikD are stronger than the ones detected for AVR-Pia and AVR1-CO39 as seen on stringent selection conditions (TDO + 10 mM 3AT). We detected all proteins fused to the GAL4 activation domain (AD) or DNA-binding domain (BD) by western blot (Supplementary Fig. 3). These results show that modification of the RGA5_HMA surface composed of β strands 2 and 3 is sufficient to confer AVR-PikD-binding and does not or only moderately affect AVR-Pia and AVR1-CO39 binding. Polymorphic residues within the m2 area seem to have limited influence on the binding of AVR-PikD to RGA5_HMA.

**Engineered RGA5_HMA domains interact strongly with AVR-PikD in vitro.** To further characterize these interactions in vitro, we performed surface plasmon resonance (SPR) experiments with recombinant HMA domains fused to the maltose-binding protein (MBP) and 6×His-tagged AVR effectors expressed in *Escherichia coli* and purified to homogeneity by affinity and size exclusion chromatography (Supplementary Fig. 4). The MBP-tagged HMA domains were captured on a chip and response units (RU) were measured following the injection of the different AVRs at 1 μM (Fig. 3). Comparison of the binding profiles revealed a tight interaction of the AVR-PikD effector with both RGA5_HMAm1 and RGA5_HMAm1m2, similar to that observed with Pikp-1_HMA, whereas poor binding was detected with wild-type RGA5_HMA (Fig. 3c, d). This was confirmed by performing successive injections of AVR-PikD at increasing concentrations (Supplementary Fig. 5). Equilibrium binding constants ($K_D$) calculated from these data indicate that the binding affinity of AVR-PikD for both RGA5_HMA mutants is in the nanomolar range as for Pikp1_HMA while its affinity for wild-type RGA5_HMA is in the μ-molar range and thus three to four

| BD | AD | DDO | TDO + 0.5 mM 3AT | TDO + 10 mM 3AT |
|---|---|---|---|---|
| AVR-Pia | RGA5_HMA<br>HMA_m1<br>HMA_m2<br>HMA_m1m2<br>RGA5_C-ter<br>C-ter_m1<br>C-ter_m2<br>C-ter_m1m2<br>Pikp-1_HMA<br>AD | | | |
| AVR1-CO39 | RGA5_HMA<br>HMA_m1<br>HMA_m2<br>HMA_m1m2<br>RGA5_C-ter<br>C-ter_m1<br>C-ter_m2<br>C-ter_m1m2<br>Pikp-1_HMA<br>AD | | | |
| AVR-PikD | RGA5_HMA<br>HMA_m1<br>HMA_m2<br>HMA_m1m2<br>RGA5_C-ter<br>C-ter_m1<br>C-ter_m2<br>C-ter_m1m2<br>Pikp-1_HMA<br>AD | | | |
| BD | RGA5_HMA<br>HMA_m1<br>HMA_m2<br>HMA_m1m2<br>RGA5_C-ter<br>C-ter_m1<br>C-ter_m2<br>C-ter_m1m2<br>Pikp-1_HMA<br>AD | | | |

**Fig. 2 Engineering of the HMA domain of RGA5 enables binding of AVR-PikD in yeast.** Interaction of AVR-PikD, AVR1-CO39, and AVR-PikD (without signal peptides and fused to the DNA binding domain (BD) of the GAL4 transcription factor) with the HMA (residues 991–1072) and C-terminal (residues 883–1116) domains of RGA5 and variants carrying mutations designed to introduce the AVR-PikD-binding surface (fused to GAL4 activation domain (AD)) was assayed by yeast two-hybrid experiments. The HMA domain of Pikp-1 (AD:HMA_Pikp-1) and the AD and BD domains alone were used as controls. Four dilutions of diploid yeast clones (1/1, 1/10, 1/100, 1/1000) were spotted on synthetic TDO medium (-Trp/-Leu/-His) supplemented with 0.5 mM and 10 mM of 3-amino-1,2,4-triazole (3AT) to assay for interactions and on synthetic DDO (-Trp/-Leu) to monitor proper growth. Pictures were taken after 5 days of growth.

orders of magnitudes lower (Supplementary Table 2). Weak yet specific binding was observed for the interaction of either AVR1-CO39 or AVR-Pia to RGA5_HMA mutants and wild type (Fig. 3a, b, d), consistent with the affinity constants in the μ molar range previously reported for the binding of RGA5_HMA for both ligands[31]. Pikp-1_HMA does not bind AVR1-CO39 but shows, as already described, weak binding to AVR-Pia[33]. The AVR1-CO39_T41G and AVR-Pia_F24S mutants that do not interact with RGA5_HMA and are not recognized by RGA4/RGA5[30,31] fail to bind any of the tested HMA domains (Fig. 3a, b).

In vitro binding is thus consistent with the yeast two-hybrid experiments and supports that engineering the β2–β3 surface of RGA5_HMA creates a strong binding to AVR-PikD. Besides,

these results demonstrate that the mutations in RGA5_HMA β strands 2, 3 and 4 do not influence the binding of AVR1-CO39 or AVR-Pia.

**Engineered RGA5_HMA domains associate with AVR-PikD in planta.** To confirm these interactions *in planta*, we performed co-immunoprecipitation (Co-IP) experiments in *Nicotiana benthamiana* using HA-tagged effectors and YFP-tagged HMA domains. Consistent with yeast two-hybrid and SPR results, AVR-PikD is co-precipitated with RGA5_HMAm1, RGA5_HMAm1m2, and Pikp-1_HMA but does not associate with RGA5_HMA (Fig. 4). AVR-Pia specifically associates with the

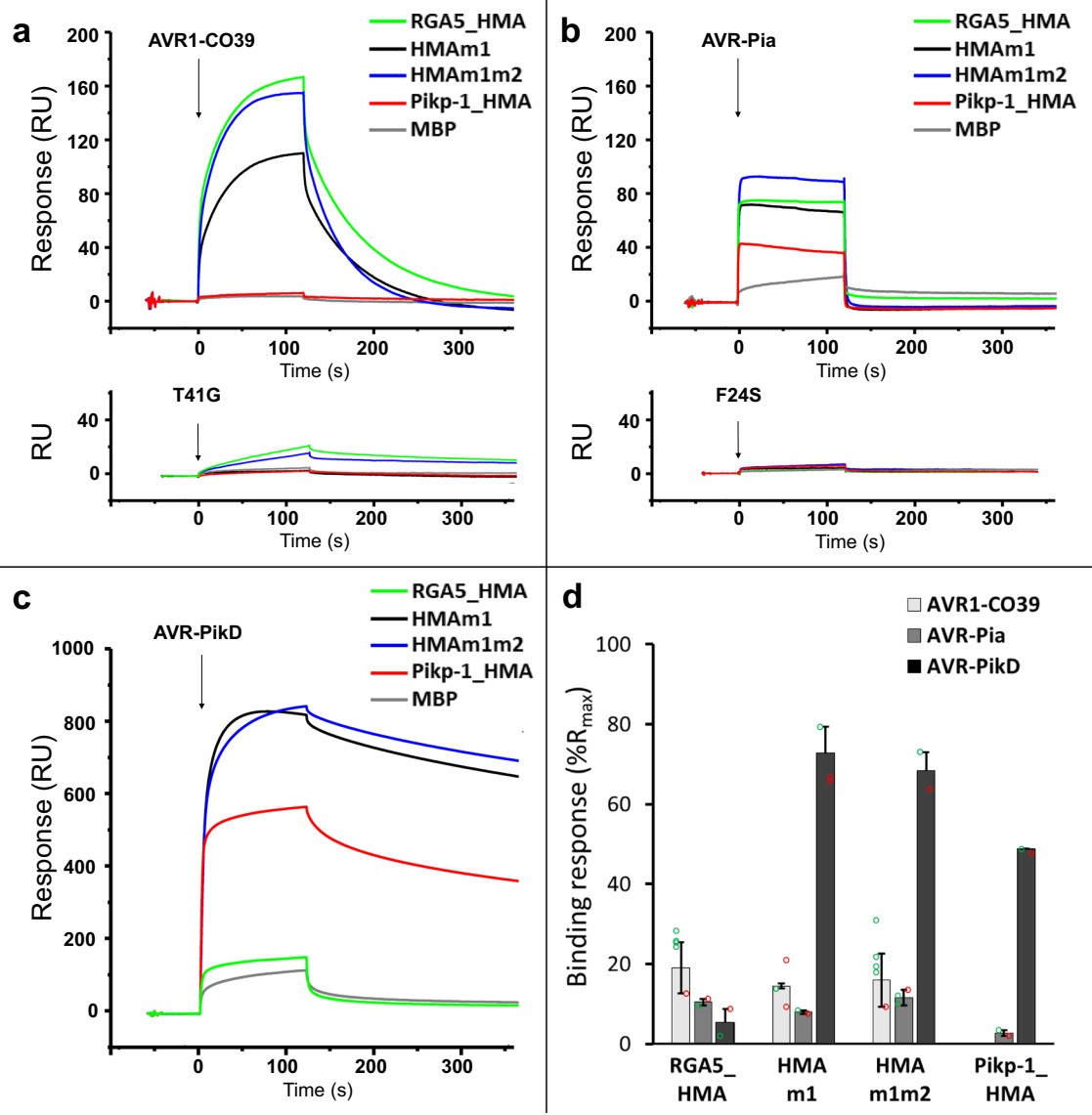

**Fig. 3 Engineered RGA5_HMAs bind strongly to AVR-PikD in vitro. a–c** AVR1-CO39 (**a**), AVR-Pia (**b**), and AVR-PikD (**c**) were injected (black arrows) at 1 µM for 2 min on the different MBP:HMA fusion proteins captured by anti-MBP antibody immobilized on the chip. Superimposed sensorgrams are shown for wild-type RGA5_HMA (green), RGA5_HMAm1 (black), RGA5_HMAm1m2 (blue), Pikp-1_HMA (red), as well as for MBP alone (gray) that serves as a negative control. The binding curves obtained with the wild-type and an inactive variant of AVR1-CO39 (**a**) or AVR-Pia (**b**) are shown in the top and lower insets, respectively. **d** Comparison of the binding response (bound fraction) at 1 µM of AVR effectors, expressed as the percentage of the theoretical maximum response (%Rmax) normalized for the amount of MBP:HMA immobilized on the chip and corrected for the MBP-tag contribution. Bars and error bars represent the mean and average deviation calculated for the %Rmax values estimated from $n = 2$ independent experiments carried out on two different days (open green and red circles) wit $n = 1$ to $n = 4$ technical replicates per experiment. Two independently purified protein samples were used to test the binding of the AVR-PikD effector to the different HMAs.

wild type, the m1 and the m1m2 HMA domains of RGA5 but does not co-precipitate with Pikp-1_HMA. We could not test AVR1-CO39:HA because it is not detected in *N. benthamiana* protein extract (Supplementary Fig. 6). Interestingly, we observed a correlation between AVR-PikD accumulation in the input and its specific associations with HMA domains suggesting that this effector protein is stabilized upon HMA-binding (Fig. 4). As expected, AVR-PikC, a non-recognized allele of AVR-Pik used as a negative control is not co-precipitated with any of the HMA domains[12].

We performed further Co-IP experiments using full-length RGA5 carrying the m1, m2, or m1m2 mutations to test the association of the entire receptors with AVR-Pia and AVR-PikD (Fig. 5). HA-tagged effectors and YFP-tagged NLRs were

transiently expressed in *N. benthamiana*. Consistent with previous results AVR-PikD is co-precipitated with RGA5m1, RGA5m1m2, and Pikp-1_HMA but does not associate with wild-type RGA5 nor with RGA5m2 (Fig. 5). AVR-Pia specifically associates with RGA5 wild type, m1, m2, and m1m2 but does not co-precipitate with Pikp-1_HMA. AVR-PikC, and PWL2, an unrelated non-MAX effector of *M. oryzae*, do not associate with any of the NLRs. To further exclude that AVR-PikD interacts with other domains of RGA5, we performed Co-IP using a YFP-tagged RGA5 construct deleted for the HMA domain (YFP:R-GA5_ΔHMA), and analyzed its association with HA-tagged AVR-PikD alongside HA-tagged AVR-Pia used as a positive control and YFP:Pikp-1_HMA serving as a control for effective and specific co-precipitation of AVR-PikD. We found that

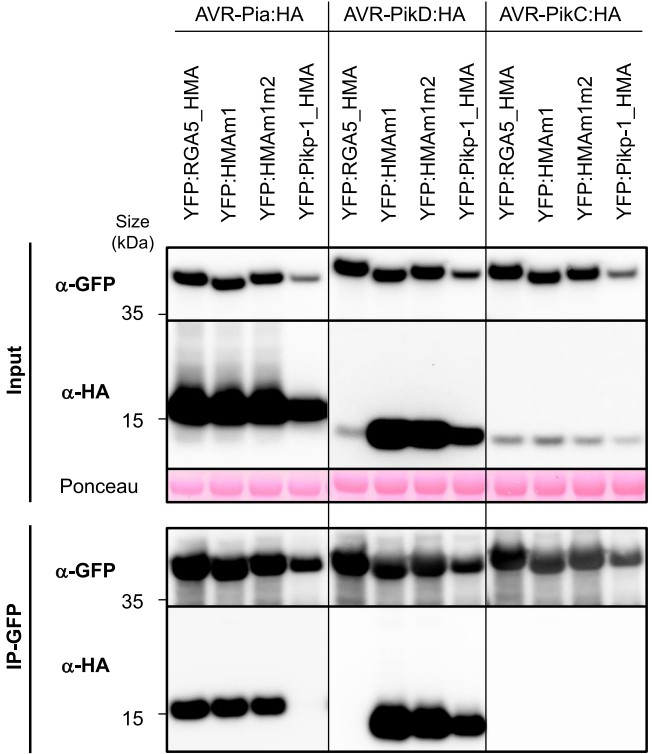

**Fig. 4 In planta association of the engineered HMA domains of RGA5 with MAX effectors.** The indicated HMA domains of RGA5 (wild type, m1, and m1m2) and Pikp-1, fused to YFP, were transiently co-expressed in *N. benthamiana* leaves with HA-tagged AVR-Pia, AVR-PikD, and AVR-PikC (without signal peptides). Proteins were extracted after 48 h, separated by gel electrophoresis, and tagged proteins were detected in the extract (input) and after immunoprecipitation with anti-GFP beads (IP-GFP, trapping YFP fusions) by immunoblotting with anti-HA (α-HA) and anti-GFP (α-GFP) antibodies. Protein loading in the input is shown by Ponceau staining of the large RuBisCO subunit. The experiment was carried out twice with similar results.

YFP:RGA5_ΔHMA co-precipitates AVR-Pia, as previously reported, but not AVR-PikD (Fig. 6)[30].

Taken together, these experiments indicate that AVR-PikD binds RGA5m1 and m1m2 with high affinity and exclusively through the mutant HMA domains but does not interact with wild-type HMA nor with the rest of the RGA5 protein.

**The engineered RGA5 variants recognize both AVR-Pia and AVR-PikD in *N. benthamiana*.** To test whether the m1, m2, and m1m2 mutations enable AVR-PikD recognition by RGA5 in a whole-plant context, we performed cell death assays using agroinfiltration of *N. benthamiana* leaves[34]. RGA4 used as a positive control induces cell death when expressed alone but is repressed by RGA5m1m2 as effectively as with wild-type RGA5 (Fig. 7). This indicates that the m1m2 mutation does not affect the functional interaction between RGA5 and RGA4. Upon co-expression of AVR-PikD and RGA4/RGA5m1m2, a cell death response is induced showing that the m1m2 mutation enables AVR-PikD recognition by RGA5 in the heterologous *N. benthamiana* system (Fig. 7). As controls, we observed that AVR-PikD is not recognized by RGA4/RGA5 but induces a strong cell death response upon co-expression with Pikp-1/Pikp-2 (Fig. 7, Supplementary Figs. 7 and 8). AVR-Pia is specifically recognized by RGA4/RGA5 and RGA4/RGA5m1m2 but not by Pikp-1/Pikp-2 (Fig. 7, Supplementary Fig. 7).

We also tested AVR-Pia and AVR-PikD recognition by RGA4 and RGA5 carrying only the m1 or m2 surfaces (Supplementary Fig. 8). RGA5m1 recognizes AVR-PikD but to a lesser extent than RGA5m1m2. The m1 mutation does not affect AVR-Pia recognition. RGA5m2 recognizes AVR-Pia but not AVR-PikD. All proteins expressed in *N. benthamiana* (besides the untagged AVR-Pia) are detected by western blotting (Fig. 5, Supplementary Fig. 6).

Consistent with the yeast-two hybrid, Co-IP, and SPR data, these results indicate that in the heterologous *N. benthamiana* system, the engineered RGA5 m1, and m1m2 efficiently bind and recognize both AVR-PikD and AVR-Pia whereas only AVR-Pia is bound and recognized by wild-type RGA5 and RGA5m2.

**The engineered RGA5 variants do not recognize AVR-PikD but retain AVR-Pia and AVR1-CO39 recognition in transgenic rice plants.** To determine whether the m1, m2, and m1m2 mutations of RGA5 confer AVR-PikD recognition in the homologous rice system, we co-transformed the rice cultivar Nipponbare (*pia⁻*/*pikp⁻*) with constructs carrying under the control of their native promoter the genomic sequences of *RGA4* and *RGA5m1*, *m2*, *m1m2*, or wild type. Transgenic rice lines transformed with *RGA4* and the *pUbi::GFP* construct were generated as controls. T0 transgenic plants were Polymerase chain reaction (PCR)-genotyped for the presence of *RGA4* and *RGA5* and all PCR products corresponding to m1, m2, and m1m2 transgenic lines were sequenced to ensure that they contain the appropriate mutations (Supplementary Table 3). This identified 14 independent T0 transgenic lines successfully transformed with *RGA4/RGA5*, 6 with *RGA4/RGA5m1*, 5 with *RGA4/RGA5m2*, 6 with *RGA4/RGA5m1m2*, and 14 with *RGA4/GFP* (Supplementary Table 3).

When possible, thalli from individual T0 plants were split and inoculated with the *M. oryzae* isolate Guy11 transformed with either the empty vector (Guy11_EV) or with *AVR-Pia* (Guy11_AVR-Pia), or with the wild-type strain JP10 naturally carrying *AVR-PikD* and lacking *AVR-Pia*. As expected, all rice transgenic lines inoculated with Guy11_EV show susceptibility (Supplementary Table 3, Supplementary Fig. 9). Upon inoculation with Guy11_AVR-Pia, 6/8 *RGA4/RGA5*, 1/2 *RGA4/RGA5m1*, 3/3 *RGA4/RGA5m2*, and 3/5 *RGA4/RGA5m1m2* T0 transgenic lines showed resistance, while all those inoculated with JP10 developed disease phenotypes. The K60 rice cultivar carrying the *Pikp-1/Pikp-2 NLR* pair and diagnostic for Pikp resistance was resistant to JP10, indicating proper AVR-PikD recognition[35,36]. These results indicate that the engineered RGA5m1, m2, and m1m2 variants all recognize AVR-Pia but not AVR-PikD in the homologous rice/*M. oryzae* system.

To back up this analysis, which was performed on T0 transgenic plants that are highly heterogeneous and stressed due to regeneration from in vitro culture, inoculation experiments were also carried out on T1 plants (Supplementary Table 3, Fig. 8). For this, we used a transgenic Guy11 isolate transformed with *AVR-PikD* instead of JP10 to ensure homogeneous fungal background. We also included the Guy11_AVR1-CO39 transgenic isolate. Based on seed availability, 3 independent rice transgenic lines were selected for *RGA4 + RGA5* and *RGA4 + RGA5m2*, 2 for *RGA4 + RGA5m1*, and 1 for *RGA4 + RGA5m1m2*. Consistent with the results obtained in the T0 experiment, AVR-Pia is recognized by RGA5 and its variants, but not AVR-PikD. In addition, like the RGA5 wild type, RGA5m1, m2, and m1m2 also recognize AVR1-CO39.

To identify a potential partial resistance to AVR-PikD in the transgenic lines, we conducted inoculation experiments using T2 plants (Supplementary Table 3, Supplementary Fig. 10) and precisely measured lesion size using the computational image

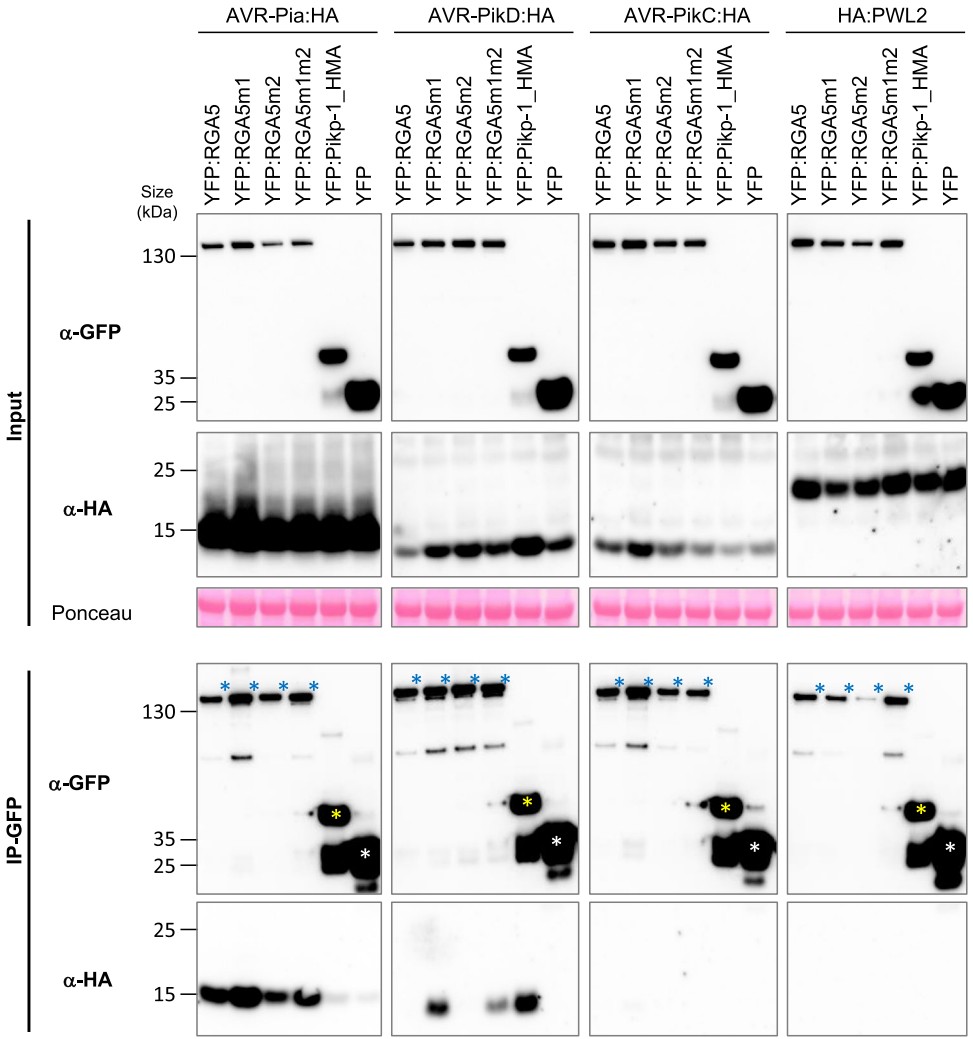

**Fig. 5 *In planta* association of the engineered full-length RGA5 receptors with MAX effectors.** The full-length RGA5 protein and related m1, m2, and m1m2 variants fused to YFP were transiently co-expressed in *N. benthamiana* leaves with the indicated HA-tagged effectors (without signal peptides). The HMA domain of Pikp-1 (fused to YFP) was used as a control as well as YFP alone and the *M. oryzae* effector PWL2. Proteins were extracted 48 h after infiltration, separated by gel electrophoresis, and tagged proteins were detected in the extract (input) and after immunoprecipitation with anti-GFP beads (IP-GFP) by immunoblotting with anti-HA (α-HA) and anti-GFP (α-GFP) antibodies. In the IP-GFP/α-GFP panel, yellow and white asterisks indicate the YFP:Pikp-1_HMA and YFP proteins respectively, while blue asterisks show the YFP:RGA5 wild type and mutant proteins. Ponceau staining shows equal protein loading in the input. The experiment was carried out twice with identical results.

analysis package Leaftool (https://github.com/sravel/LeAFtool). These experiments show that in the transgenic RGA5m1, m2, and m1m2 plants disease lesions caused by Guy11_AVR-PikD are not reduced in size compared to the control rice lines or to infection with the Guy11_EV isolate. However, AVR-Pia induces resistance manifested by small hypersensitive response-type lesions equally in all RGA5 lines (Supplementary Fig. 10). This further confirms that, in rice, AVR-PikD is not recognized by the engineered RGA5 variants.

To verify the expression of the *RGA4* and *RGA5* transgenes, we performed qRT-PCR experiments on the transgenic T2 lines used in the inoculation experiments. They show that both transgenes are properly expressed, which is consistent with the Pia and Pi-CO39 resistance phenotype in all RGA4/RGA5 lines (Supplementary Fig. 11).

## Discussion

NLR immune receptors provide plants with efficient protection from biotrophic pathogens and the corresponding genes are

therefore critical for breeding disease-resistant crops. However, not all NLR resistance genes are useful for crop protection since their spectrum can be extremely limited due to the polymorphism of effector-coding genes. In certain cases, no NLR inducing broad-spectrum resistance is present in crop germplasm. A promising prospect to overcome these limitations is creating NLRs with specific recognition specificities by molecular engineering[3]. NLR-IDs have been recognized as prime candidates for such resistance engineering due to their modular structure and the well-established role of decoy IDs in effector recognition[11,13,14].

In the RGA4/RGA5 and Pik1/Pik2 model systems, detailed structure–function analysis deciphered at atomic scale how IDs contribute to the recognition of fungal effector proteins[12,30–32]. These studies established that small HMA proteins have been recruited repeatedly and independently to serve as decoy domains that physically bind effectors like molecular traps[15,16]. These breakthroughs pave the way toward rational structure-guided design of effector-binding domains in these NLRs. The first proof of concept studies in the Pik-1/Pik-2 system demonstrated how structure-guided ID engineering enabled the recognition of

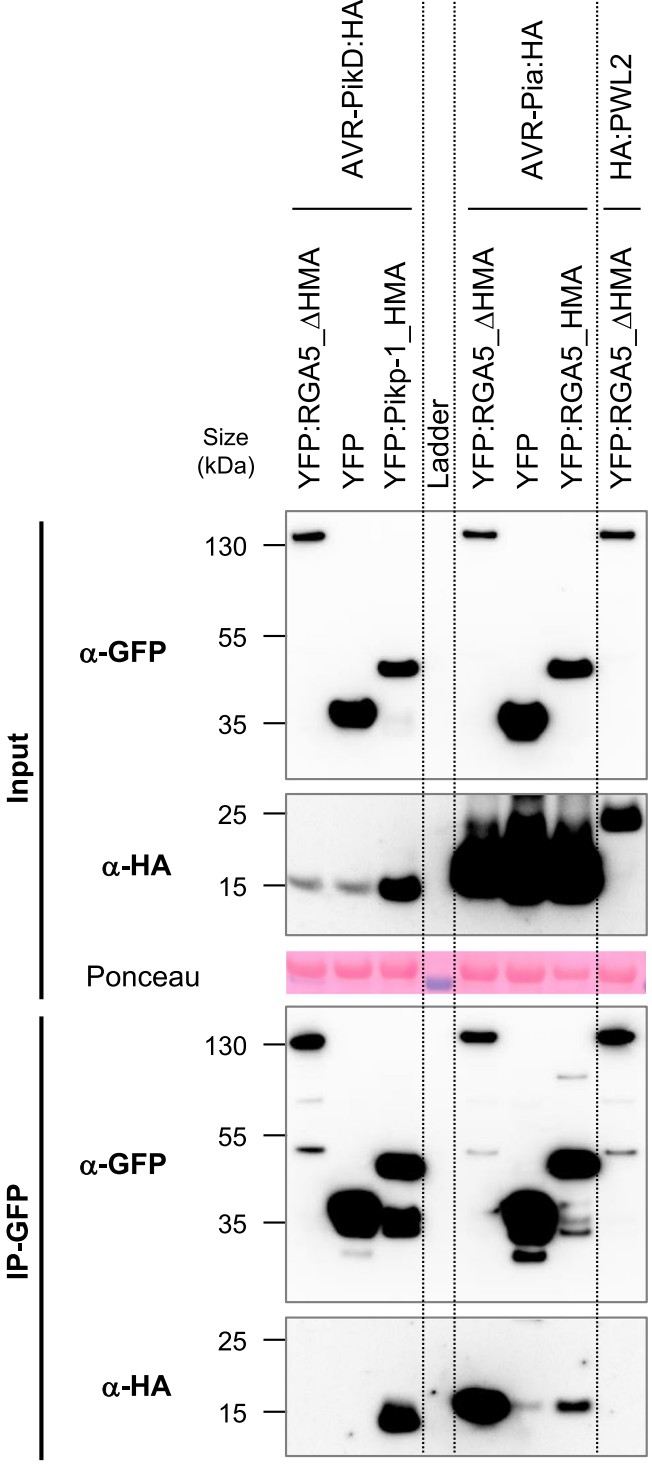

**Fig. 6 AVR-Pia associates with RGA5_ΔHMA but not AVR-PikD.** RGA5 deleted of its C-terminal domain (RGA5_ΔHMA, residues 1–996) fused to YFP was transiently co-expressed in *N. benthamiana* leaves with HA-tagged AVR-Pia, AVR-PikD, or PWL2 (without signal peptides). Proteins were extracted after 48 h and tagged proteins were detected in the extract (input) and after immunoprecipitation with anti-GFP beads (IP-GFP) by immunoblotting with anti-HA (α-HA) and anti-GFP (α-GFP) antibodies. Protein loading in the input is shown by Ponceau staining of the large RuBisCO subunit. The HMA domains of Pikp-1 and RGA5 were used as controls as well as the unrelated PWL2 effector of *M. oryzae*. The experiment was carried out twice with identical results.

different alleles of the same effector[26]. However, the creation of entirely different specificities by ID engineering has not been reported yet.

In this study, we successfully generated a new effector recognition specificity for the RGA5 immune receptor by creating a novel binding surface in its HMA ID. For this, we swapped a limited number of residues from the Pikp-1_HMA into the RGA5_HMA domain. The identification of critical residues to be mutated relied on structural knowledge obtained at high resolution for these HMA domains in complex with different AVRs[12,31,32]. The resulting mutants, RGA5_HMAm1 and RGA5_HMAm1m2, were assessed in silico for effector binding by modeling of relevant effector/HMA complexes. These models predicted that both RGA5_HMA mutants could form complexes with AVR1-CO39, AVR-Pia, and AVR-PikD and that binding affinity in these complexes would be similar to those in the respective complexes formed by wild-type RGA5_HMA and Pikp1_HMA. In vivo and in vitro experiments confirmed these predictions and demonstrated that RGA5_HMAm1 and RGA5_HMAm1m2 bind AVR-PikD with high affinity in addition to retaining moderate binding affinity to AVR-Pia and AVR1-CO39.

The $K_D$ values retrieved from our SPR experiments indicated nano-molar affinity (ranging from 0.5 to 1.6 nM, Supplementary Table 2) for the complexes formed between AVR-PikD and the engineered RGA5_HMAs, similar to that estimated for the complex formed with the wild type Pikp-1_HMA. This is consistent with the $K_D$ value of 6 nM formerly determined by SPR for AVR-PikD/Pikp-1_HMA interaction using different constructs and experimental settings[32]. Therefore, our results with the RGA5_HMA mutants further illustrate the previously established critical role of the HMA β2 and β3 strands in AVR-PikD binding and recognition[12,32]. The fact that m2 mutations have a low impact on AVR-PikD binding to RGA5_HMA is also consistent with the limited contribution of the β4 strand of Pikp-1_HMA to AVR-PikD specific recognition[26]. However, this β4 strand is important for the specificity of recognition of other AVR-Pik alleles by Pik-1 alleles[26,32,33]. In contrast, the m1 and m2 mutations did not affect the binding of the naturally recognized effectors AVR-Pia and AVR1-CO39, which both bind with low affinity to the α1/β2 surface located on the other side of the HMA domain (Fig. 1b and Supplementary Fig. 1B). The affinity constants retrieved from SPR data for the interaction of RGA5_HMAm1m2 with AVR-Pia (5.9 µM) or AVR1-CO39 (6.3 µM) are very similar to those we estimated for the wild-type in the present study (7.2 and 4.9 µM) or in previous ITC measurements (7.8 and 7.2 µM)[30,31]. This weak impact of the m1 and m2 mutations on AVR-Pia and AVR1-CO39 binding further confirms the current structural model for the detection of these effectors and nicely illustrates the fact that different MAX effectors can bind two different interfaces in the HMA domain. In addition, it shows that the decoy domain of an NLR can be engineered to create a high-affinity binding site for a novel effector without affecting the binding of naturally recognized effectors.

Previous work established the central role of HMA binding in the detection of pathogen effectors by the RGA5 and Pikp-1 immune receptors[12,30,31]. Indeed, effector mutants that are strongly impaired in binding to the HMA lose their avirulence activity and do no longer trigger immunity in resistant rice cultivars. The RGA5m1 and m1m2 mutants created in this study gain high-affinity binding to AVR-PikD and activate cell death immune responses when co-expressed with this effector in the heterologous *N. benthamiana* model system. This indicates that the new-to-nature receptors harboring the engineered HMA

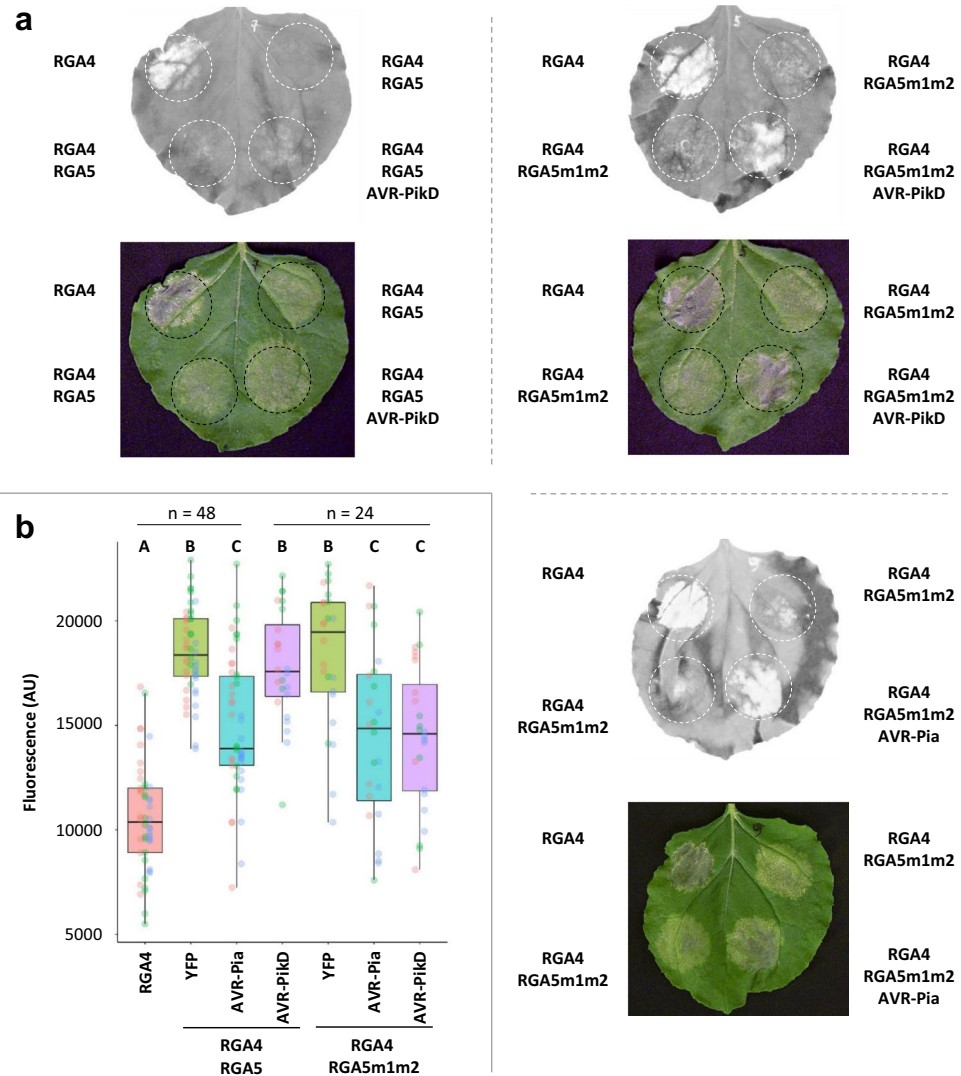

**Fig. 7 The RGA5m1m2 mutant recognizes AVR-Pia and AVR-PikD in *N. benthamiana*. a** The indicated combinations of constructs were transiently expressed in *N. benthamiana* leaves. The AVR-PikD (without signal peptide) and RGA4 constructs carry an HA tag at their C-termini, while RGA5 and RGA5m1m2 are YFP-tagged at their N-termini and AVR-Pia (without signal peptide) is untagged. The auto-active RGA4 construct was used as a positive control for cell death induction. Cell death was visualized 5 days after infiltration. Greyscale pictures were taken using a fluorescence scanner with settings allowing visualization of the disappearance of red fluorescence due to cell death (white patches of dead cells). A picture of corresponding leaves is shown below. **b** Cell death was quantified by measuring fluorescence levels (arbitrary unit, AU) in the infiltrated areas using ImageJ[34]. The resulting data were plotted. The boxes represent the first quartile, median, and third quartile. The difference in fluorescence levels was assessed by an ANOVA followed by a Tukey HSD test. Groups with the same letter (**a–c**) are not significantly different at level 0.05. For each combination of constructs, all of the measurements are represented as dots with a distinct color (red, green, and blue) for each of the three biological replicates.

domains are capable of forming functional assemblies responding to AVR-PikD in addition to the naturally recognized effectors AVR-Pia and AVR1-CO39. However, the cell death response is similar or slightly weaker than with AVR-Pia although the binding of RGA5_HMAm1m2 to AVR-PikD detected in yeast or by SPR is much stronger than to AVR-Pia and AVR1-CO39. Similar discrepancies between the gain-of-binding observed in vitro and the eliciting of the cell-death response in *N. benthamiana* have also been reported for a Pikp-1 variant with an engineered HMA domain[26]. Therefore, for RGA5 as for Pikp-1, the effector/HMA binding affinity does not necessarily correlate with the strength of the induced immune response. This suggests that additional factors may influence effector recognition by these NLRs.

When transformed into rice, RGA5m1 and RGA5m1m2 do not confer resistance to *M. oryzae* isolates carrying *AVR-PikD*

although they are as active as wild-type RGA5 in recognizing *AVR1-CO39* and *AVR-Pia* isolates. This indicates that in the homologous rice system the binding of an effector to the HMA domain even with high affinity is not sufficient to activate the RGA4/RGA5 complex and to trigger immune responses. Two main hypotheses can explain this result and the missing correlation between HMA ID binding affinity and response in *N. benthamiana* cell death assays. HMA-binding alone may not be sufficient for efficient receptor activation because additional interactions with RGA5 outside the HMA domain are required for full receptor activation. Support for this hypothesis comes from the finding that AVR-Pia associates with RGA5_ΔHMA in Co-IP experiments while AVR-PikD does not. Such additional interactions between the effector and RGA5 outside of the HMA domain should not only stabilize effector binding and significantly increase overall binding affinity but may directly induce

**Fig. 8 The RGA5 m1, m2, and m1m2 mutants recognize AVR-Pia but not AVR-PikD in rice.** The rice cultivar Nipponbare was co-transformed with a genomic construct for *RGA4* and a genomic construct for *RGA5*, *RGA5m1*, *RGA5m2*, or *RGA5m1m2*. A transgenic line carrying *RGA4* and *GFP* was also generated as a control. T1 plants of the transgenic lines were spray inoculated with the transgenic strains Guy11-AVR-Pia, Guy11-AVR1-CO39, Guy11-AVR-PikD, or Guy11-EV. The rice cultivar K60 carrying the *Pikp* resistance was used as a control for AVR-PikD specific recognition. Pictures show representative symptoms at 7 days after inoculation. Individual leaves indicate independent T1 transgenic lines. Similar results were obtained in two independent inoculation experiments performed on T0 (Supplementary Fig. 8) and T2 (Supplementary Fig. 9) plants.

conformational changes in RGA5 required for efficient RGA4/RGA5 receptor complex activation[30]. Additional support for this hypothesis comes from the previous investigation of the recognition of AVR-Pia and AVR1-CO39 mutants by RGA4/RGA5 that showed high resilience to reduction of HMA-effector affinity[30,31]. Indeed, effector mutants with strongly reduced HMA affinity were still recognized and only complete loss of HMA-binding resulted in the loss of avirulence activity. Interestingly, the recently reported structures of two different resistosome composed of the TIR-NLRs ROQ1 or RPP1 in complex with their matching effectors (XopQ and ATR1, respectively) show examples where effectors recognition relies on direct binding to multiple sites and domains of the NLRs[37,38]. In both cases, the effectors do not only bind an extended surface of the LRR domain but also the post-LRR domain that adopts a jelly roll/Ig-like fold, which occurs specifically and with high frequency in TNLs and is important for receptor function[39,40].

Alternatively, effectors may have to bind to the proper side of the RGA5_HMA domain to trigger activation of RGA4/RGA5. This may be due to steric or spatial constraints in the receptor complex or because effectors have to disrupt intramolecular interactions mediated by the AVR-Pia and AVR1-CO39-binding surface of RGA5_HMA. The AVR-Pia and AVR1-CO39-binding surface of RGA5_HMA could also bind rice proteins not present in *N. benthamiana* that have to be displaced by the effectors for proper activation of the immune receptor complex. These alternative hypotheses are supported by a recent report on the engineering of RGA5 for recognition of AvrPib, another MAX effector from *M. oryzae*[41]. The study shows that mutations in the

HMA domain at residues of the AVR1-CO39-binding surface (α1 and β2) as well as proximal C-terminal residues confer AvrPib binding, recognition, and immunity in rice. Co-IP data show that AvrPib does not bind to wild-type RGA5 indicating that this effector does not associate with any domain of this NLR[41]. The affinity of AvrPib to the engineered RGA5 HMA domain ($K_D$ of 150 µM) is much lower than that of AVR-Pia to the wild-type HMA domain of RGA5. This suggests that weak effector binding is sufficient for recognition as long as it involves the α1 and β2 surface. Liu et al.[41] also report that their engineered RGA5 loses the ability to recognize AVR-Pia, revealing the difficulty to confer multiple effector recognition specificities to this NLR-ID receptor.

The recognition of AVR-PikD by RGA5m1 and m1m2 in *N. benthamiana*, despite a potentially incomplete receptor binding, may be due to the strong overexpression of effectors and receptors in this system. The resulting high cellular levels of receptors and effectors may overcome sub-optimal conditions and enable to reach the threshold that is required for activated receptor complexes triggering immune responses and cell death. Similar discrepancies between effector recognition in *N. benthamiana* cell death assays and pathogen resistance in the homologous system have been reported in a study that aimed to engineer the NLR R3a from potato for recognition of additional alleles of the effector AVR3a from *Phytophthora infestans*[23].

While NLRs from many different clades can carry IDs, the frequency of NLR-IDs in most NLR clades is low. In cereals, only 3 NLR clades named major integration clades (MIC) are strongly enriched for NLR-IDs[9]. In MIC2 and MIC3, IDs are conserved

(DDE superfamily endonuclease and the BED-type zinc finger domains, respectively), indicating that they originate from unique ancient integration events. Only MIC1, to which belongs to RGA5, harbors highly diverse IDs that correspond to many different types of protein domains. Pikp-1 belongs to another NLR clade that shows much less frequent ID integration. Comparison of orthologous MIC1 NLRs indicates frequent and ongoing exchange of IDs[5,9]. Since MIC1 NLRs can accommodate such a huge diversity of IDs and since swap-in of novel IDs seems common mechanisms and a frequent event in their evolution, they appear particularly suited "chassis" for generating "à la carte" novel effector-recognition specificities by molecular engineering and ID manipulation. Until recently, such engineering was hampered by the lack of knowledge regarding the molecular mechanisms of NLR receptor action, both in the specific recognition of effectors and in their underlying activation. Major breakthroughs in the determination of the 3D structure of effectors and NLR receptors are lifting these barriers and paving the way toward structure-guided engineering of NLR receptors.

Based on structural knowledge on effector-ID interactions, we succeeded in generating artificial NLR receptors that bind a novel effector through a previously not-existing binding interface and have a novel recognition specificity in heterologous *N. benthamiana* cell death assays. However, the finding that these engineered NLRs do not provide a novel resistance specificity in the homologous rice system illustrates our limited understanding of the action of IDs in full-length receptors and of the interactions in NLR pairs at the molecular level. In particular, our finding that high-affinity binding of a new effector to the HMA β3 β4 interface does not confer immunity while low-affinity binding of another effector to the α1 β2 interface does[41] is intriguing and suggests that proper spatial positioning of the effector is important. In the case of AVR-Pia, additional interactions with RGA5 outside the ID might be important for stabilizing the interaction and for recognition. Further analyses will be crucial to unravel the molecular mechanisms of effector recognition by NLR-IDs.

In addition, it is critical to understand more precisely how the binding of effectors to RGA5 de-represses the RGA4/RGA5 complex. The recent description of the molecular and structural mechanism of effector-dependent activation of the Arabidopsis CNL ZAR1 provides a blueprint for the understanding of CNLs that act alone and rely on decoy or guard proteins for effector recognition[42,43]. Similarly, detailed studies are required to provide a molecular and structural framework for MIC1 NLR action and their interaction with RGA4-like executor NLRs. It is tempting to speculate that RGA4/RGA5 may form upon activation a similar multimeric resistosome complex as ZAR1/RKS1[44] and form, at the inactive state, a repressed heterodimeric complex. However, to address this hypothesis and to decipher the intra and intermolecular interactions in these complexes and the conformational changes mediating their transitions will require an integrated multidisciplinary approach combining structure biology, in vitro biochemistry, and functional plant genetics. In addition, it would be highly instrumental to dispose of additional MIC1 NLR model systems where the effectors and their interaction with the ID of the RGA5-like sensor are known to decipher the commonalities and the specificities in different MIC1 NLR pairs. In this sense, this study not only illustrates the potential that NLR-IDs provide for creating an almost unlimited number of pathogen recognition specificities but also the tremendous challenges that remain to be addressed before this ambitious goal can be reached.

## Methods

**Protein expression and purification.** Details on the plasmid constructs and synthetic genes used in the present study are given in Supplementary Table 5. Wild-type and mutant AVR1-CO39 and AVR-Pia proteins, deleted for their endogenous secretion signal, were expressed and purified as His-tagged proteins from *Escherichia coli* periplasm[29]. Bacteria were incubated for 30 min in lysis buffer (200 mM Tris-HCl pH8, 200 mM Sucrose, 0.05 mM EDTA, 50 μM lysozyme), debris was removed by centrifugation at 12,000$g$ for 15 min at 4 °C. His-tagged protein was purified from the crude protein extract using an AKTA basic system with a HisTrap 5 ml HP column (GE Healthcare), equilibrated in buffer A (50 mM Tris-HCl, pH 8.0, 300 mM NaCl, 1 mMDTT, 0.1 mM Benzamidine), and elution with buffer B (buffer A supplemented with 500 mM imidazole). Fractions containing the target protein according to sodium dodecyl sulfate-polyacrylamide gel electrophoresis (SDS-PAGE) were pooled and used for gel filtration with a Superdex S75 26/60 (GE Healthcare) column in buffer A for further purification. For AVR-PikD, the His-tagged protein was expressed in *E. coli* BL21 (DE3) from the pET derived plasmid pdbccdb_3C_His (kindly provided by F. Allemand) and purified under denaturing conditions by affinity chromatography (50 mM Tris, pH 8.5, 300 mM NaCl, 1 mM DTT, 8 M urea), refolded by over-night dialysis and further purified by gel filtration in 20 mM Tris, pH 8.5, 150 mM NaCl, 1 mM DTT. Wild-type and mutant HMA domains from RGA5 and Pikp-1 fused to the maltose-binding protein (MBP) were expressed from a pdbccdb_3C_His_MBP construct and purified on His-trap column followed by over-night dialysis in 20 mM Tris pH 7.5, 150 mM NaCl, 1 mM DTT. Protein concentration was determined by absorption of UV light at 280 nm using a NanoDrop and theoretical extinction coefficients. All protein samples were stored at −20 °C.

**Surface plasmon resonance.** SPR experiments were performed at 25 °C on a Biacore T200 apparatus (GE Healthcare) in running buffer (20 mM Tris, pH 8, 150 mM NaCl, 1 mM DTT) supplemented with 0.05% of P20 surfactant (GE Healthcare). Anti-MBP murine monoclonal antibody (Biolabs) was immobilized (around 11000 RU) on the CM5S dextran sensor chip (Cytiva) by amine coupling according to the manufacturer's instructions. The MBP protein alone or fused to the wild-type or mutant HMA domain from RGA5 or Pikp-1 was then injected at 300 nM on the sensor chip, leading to a capture level of 3000–4000 RU. For comparative binding experiments, the concentrated AVR effectors were diluted at 1 μM in running buffer and injected at 30 μL/min during 120 s followed by a dissociation phase in flow buffer. The cycle was ended by injecting 7 μL of regeneration buffer (10 mM glycine-HCL, pH 2). All curves were analyzed using Biacore T200 BiaEvaluation software 3.0 (GE Healthcare), after double–referencing subtraction. Binding levels (bound fraction) were compared by calculating the fraction of immobilized MBP:HMA with bound AVR at the end of injection expressed as a percentage of the theoretical maximum response (%Rmax) and corrected for the contribution of the MBP tag by subtracting the bound fraction calculated for the MBP protein alone. For kinetic titration experiments (single-cycle kinetics), the analyte (AVR-PikD) was dialyzed in running buffer (20 mM Tris, pH 8, 150 mM NaCl, 0.5 mM DTT), and increasing concentrations were injected successively on captured MBP:HMAs (about 4000 RU) at a flow rate of 30 μL/min for 60 s followed by a dissociation phase of 600 s after the final injection. Data analysis and determination of binding parameters were performed with BiaEvaluation using steady-state or heterogeneous binding fitting models to obtain the best fitting.

**Growth of plants and fungi and infection assays.** Rice plants (*Oryza sativa* L.) were grown in a glasshouse under a day-time temperature of 27 °C, a night-time temperature of 22 °C, and 60% humidity. Nitrogen fertilization with 8.6 g of nitrogen equivalent was applied 7 days and 2 days before inoculation[45]. *N. benthamiana* plants were grown in a growth chamber at 22 °C with a 16 h light period. *M. oryzae* isolates and transgenic strains were grown for 7–9 days on rice flour agar medium (20 g l[−1] rice seed flour, 2.5 g l[−1] yeast extract, 1.5% agar) with a 12-h photoperiod at 26 °C[46]. For the determination of interaction phenotypes and gene expression, a suspension of fungal conidiospores (50,000 spores ml[−1]) was spray-inoculated on the leaves of 3-week-old rice plants. Rice leaves were collected and scanned 7 days after inoculation.

**Constructs for yeast two-hybrid, Co-IP, and rice transformation.** PCR products used for cloning were generated using Phusion High-Fidelity DNA Polymerase (Thermo Fisher) using primers listed in Supplementary Table 4. Details of constructs are given in Supplementary Table 5. Briefly, all ENTRY vectors used for LR cloning were obtained either by gateway BP cloning (Life Technologies) into the pDONR207 vector or through site-directed mutagenesis (Quikchange lightning technology, Agilent technologies) using an ENTRY clone as a template. Plasmids used for yeast two-hybrid or Co-IP were generated by gateway LR cloning (Life Technologies) using the ENTRY vectors described above and appropriate destination vectors listed in Supplementary Table 5. Plasmids used for rice transformation were created by site-directed mutagenesis (Quikchange lightning technology, Agilent technologies) to introduce point mutations in the genomic sequence of RGA5 already cloned in pAHC17. The resulting constructs were digested using the HindIII and BamHI restriction enzymes (BioLabs) and cloned in the pCambia2300 vector.

**Yeast two-hybrid analysis.** Yeast two-hybrid assays were performed according to the Matchmaker Gold yeast two-hybrid system protocol (Clontech). BD- and AD-fused constructs were transformed into the yeast strains Gold and Y187, respectively. Following mating, diploid yeasts were plated on synthetic DDO (-Trp/-Leu)

and TDO (-Trp/-Leu/-His) medium (supplemented with various concentrations of 3-amino-1,2,4-triazole) and incubated at 28 ℃ for 5 days.

**Transient protein expression in *N. benthamiana*.** For *N. benthamiana* transient leaf transformations, all constructs were transformed into *Agrobacterium tumefaciens* strain GV3101_pMP90. Bacterial strains were grown at 28 ℃ for 24 h in LB liquid medium containing 50 µg ml⁻¹ rifampicin, 15 µg ml⁻¹ gentamycin, and 25 µg ml⁻¹ kanamycin. Bacteria were harvested by centrifugation, resuspended in infiltration medium (10 mM MES pH 5.6, 10 mM MgCl₂, and 150 µM acetosyringone) to an OD600 of 1, and incubated for 2 h at room temperature before infiltration. The P19 suppressor of gene silencing was used in all infiltration assays at a final OD₆₀₀ of 0.1. The infiltrated plants were incubated in growth chambers under controlled conditions for all following assays[11]. Documentation of cell death was performed as described by Xi et al.[34]. Briefly, pictures of the detached leaves were taken 5 days after infiltration, and leaves were then scanned using the Typhoon FLA9000 laser scanner (GE Healthcare). The reading mode of the scanner was set to fluorescence and leaves were scanned with the 635 nm laser diode for excitation and the long-pass red filter module to collect the red fluorescence. For data acquisition, the photomultiplier tube was set to 500 V and the pixel size to 200 µm. The red fluorescence of the infiltrated leaf areas was measured with ImageJ software by quantifying the mean gray value within each area. Boxplots were generated using R v4.0.2 and the package tidyverse[47]. The difference of red fluorescence induced by the various agro-infiltrated constructs was assessed either by a one-way ANOVA followed by a Tukey HSD test or by a Kruskal Wallis test followed by a Dunn test.

**Transgenic rice lines.** *pRGA5:RGA4, pRGA5:RGA5, pRGA5:RGA5m1, pRGA5:RGA5m2, pRGA5:RGA5m1m2* and *pUBI:GFP* were used for *A. tumefaciens*–mediated transformation (strain EHa105) of wild-type Nipponbare rice[48]. Infected calli were selected on medium containing 200 mg L⁻¹ geneticin and 50 mg L⁻¹ hygromycin. Resistant calli were transferred to the regeneration medium. T0 plants were used for *M. oryzae* inoculation assays 3 weeks after transfer to soil. For this, regenerated plants with at least three tillers were split into several plantlets and replanted in the soil in independent pots. The presence of the transgenes in T0 plants was verified by PCR. Sequencing was used to confirm the identity of the m1, m2, and m1m2 constructs.

**Protein extraction immunoblot and Co-IP.** Protein extracts of *N. benthamiana* leaves were prepared in protein extraction buffer (25 mM Tris-HCl pH 7.5, 150 mM NaCl, 1 mM EDTA, 10 mM DTT, 1 mM PMSF, 0.1% NP40, 0.5% PVPP, 1% Sigma protease inhibitor and 1 tablet of Roche complete EDTA-free protease inhibitor cocktail for 50 ml of buffer). For anti-GFP immunoprecipitations, 7 µl of magnetic GFP-trap_M beads (Chromotek), prewashed three times in protein extraction buffer, were added to 1 ml of protein extract and incubated with gentle rotation for 2 h at 4 ℃. Beads were magnetically separated and washed three times with 600 µl of protein extraction buffer (without PVPP). Bound proteins were eluted by boiling for 10 min in 40 µl of Nupage LDS sample buffer with a reducing agent (Life Technologies). For Co-IP assays using full-length RGA5 (and its variants), a high stringency buffer was used for the extraction (50 mM Tris-HCl pH 7.5, 150 mM NaCl, 1 mM EDTA, 10 mM DTT, 1% NP40, 0.1% SDS, 0.5% Deoxycholate, 0.5% PVPP, 1% protease Sigma inhibitor and 1 tablet of Roche complete EDTA-free protease inhibitor cocktail for 50 ml of buffer) and the wash steps (same as in the extraction buffer without DTT, PVPP, and Sigma protease inhibitor). Total yeast protein extraction was performed using the post-alkaline extraction method (Kushinov 2000). Yeast cells from 200 µl of saturated liquid culture were harvested by centrifugation and resuspended in 100 µl distilled water. Totally, 100 µl of 0.2 M NaOH were added and cells were incubated for 5 min at room temperature, pelleted, resuspended in 50 µl of Nupage LDS sample buffer with a reducing agent (Life Technologies), and boiled for 10 min. For immunoblotting analysis, proteins were separated by SDS-PAGE using precast Bis-Tris NuPAGE gels (Life Technologies) and transferred to a nitrocellulose membrane (iBlot 2 transfer stacks, Life Technologies). For immunoblotting assays involving full-length RGA5 or RGA5_ΔHMA, proteins were wet-transferred to a nitrocellulose membrane (Millipore). Membranes were blocked in 5% skimmed milk and probed with anti-HA (Roche, anti-HA-peroxidase high affinity from rat IgG₁, clone BMG 3F10, dilution 1/1000), anti-Myc (Roche, anti-c-myc-peroxidase mouse monoclonal antibody, clone 9E10, dilution 1/1000) or anti-GFP antibodies (Roche, anti-GFP from mouse IgG₁K, clones 7.1 and 13.1, dilution 1/1000) followed, if required, by anti-mouse antibodies conjugated with horseradish peroxidase (Sigma, anti-mouse IgG peroxidase antibody produced in goat, A4416, dilution 1/10,000). Labeling was detected using the Immobilon western kit (Millipore) or the SuperSignal West Femto Maximum sensitivity substrate (Thermo Fisher Scientific). Membranes were stained with Ponceau S to confirm equal loading.

**RNA extraction from rice plants and qRT-PCR analysis.** RNA was extracted from rice leaves with Trizol reagent (Invitrogen). Reverse transcription was performed with oligo(dT)18 primers, and quantitative PCR was performed using LC 480 SYBR Green I Master mix (Roche) and a Lightcycler 480 instrument (Roche). As reference, a fragment of the rice actin gene (*Os03g50890*) was amplified. The primers used for qRT-PCR are listed in Supplemental Table 4. The primer pairs oCS429/oCS430 and oCS433/oCS434 were used for *RGA5* and *RGA4* respectively, while the 33F/33R primers were used for the rice actin gene. The Ct values were determined with the Lightcycler 480 Software (Roche, version 1.5) using the advanced relative

quantification method with the following options: Abs Quant/2nd derivative Max with high confidence. Primer efficiencies for each target and for the reference gene were calculated using a serial dilution of cDNA from the rice cultivar Kitaake (carrying *RGA4* and *RGA5*). Relative expression of *RGA4* and *RGA5* in each transgenic line was normalized relative to the mean expression level of each gene in Kitaake.

**Statistical analysis for *M. oryzae* inoculation test.** For each tested *M. oryzae* strain, lesion surfaces were measured on the youngest fully expanded leaf of 5–8 plants per independent transgenic rice line using LeAFtool (https://github.com/sravel/LeAFtool). To determine whether lesion areas on different transgenic lines of rice are significantly different, a Kruskal–Wallis test was performed followed by a Dunn test.

**Molecular modeling of HMA/AVR complexes.** The 3D model of the RGA5_HMA/AVR-Pia complex was generated by replacing the AVR1-CO39 molecule in the RGA5_HMA/AVR1_CO39 crystal structure (PDB 5ZNG) with the superimposed AVR-Pia molecule from the Pikp-1_HMA/AVR-Pia crystal structure (PDB 6Q76). The RGA5_HMAm1m2/AVR1-CO39 3D model was built by replacing in PDB 5ZNG the peptide fragments containing strands β2–β3 (Gly1024-Gly1042) and β4 (Ala1061-Val1069) by the corresponding superimposed peptide fragments from the Pikp-1 HMA molecule in PDB 6G10 (Gly215-Gly233 and Ala252-Lys262, respectively) and substituting Ala260 and Asn261 with Val and Glu in order to match the RGA5_HMAm1m2 C-terminal sequence. The AVR1-CO39 molecule was then replaced by the superimposed AVR-PikD effector molecule in PDB 6G10 in order to model the RGA5_HMAm1m2/AVR-PikD complex. The RGA5_HMAm1m2/AVR-Pia complex was modeled by replacing the effector and peptide fragments from PDB 6G10 in the RGA5_HMAm1m2/AVR-PikD model by their structural counterparts in PDB 6Q76. All models were then refined in explicit water with Charmm in Charmm36 force field[49]. The refinement protocol consisted of 100 energy minimization steps followed by 125 ps (1,25,000 steps of 1fsec) molecular dynamics at 303 K, sufficient to reach stable equilibrium as observed by reporting room mean square fluctuations of energies and temperature. Analysis of the HMA/AVR complex interface was performed with QtPISA[50].

**Reporting summary.** Further information on research design is available in the Nature Research Reporting Summary linked to this article.

## Data availability

Source data are provided with this paper. Additional datasets generated during and/or analyzed during the current study are available from the corresponding authors on reasonable request.

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

## Acknowledgements

We thank Mark Banfield and Hannah Langlands for providing the *AVR-PikD:HA* and *Pikp-1:Flag/Pikp-2:HA* constructs. We thank Corinne Michel, Aurélie Ducasse and Isabelle Meusnier for technical assistance. This research was funded by the ANR project Immunereceptor (ANR-15-CE20-0007) and benefited from the PhD fellowship of Y.X. from China Scholarship Council (CSC grant 201806350131) and from interactions promoted by COST Action Sustain FA1208 (https://www.cost-sustain.org). The CBS is a member of the French Infrastructure for Integrated Structural Biology (FRISBI), supported by the National Research Agency (ANR-10-INBS-05), and is a GIS-IBIsA platform.

## Author contributions

S.C., A.P., and T.K. designed the research. S.C., Y.X., N.D., V.C., L.M., M.P., C.H., K.d.G., and A.P. performed the research. S.C., Y.X., N.D., K.d.G., V.C., A.P., and T.K. analyzed the data. S.C., N.D., and T.K. wrote the paper.

## Competing interests

The authors declare no competing interests.
