## [Peer Review File · Nature Communications]

New effector recognition specificity in a plant immune receptor by molecular engineering of its integrated domainREVIEWER COMMENTS

Reviewer #1 (Remarks to the Author):

The manuscript authored by Cesari and colleagues presents the extensive body of work describing their attempts to expand NLR recognition specificity through a structure-guided engineering of integrated decoy domain (ID). Using the well-studied HMA ID and its cognate binding AVR partners, the authors demonstrated that new binding affinity to a specific effector could be achieved through structure-guided substitutions. The expansion of physical affinity was correlated with recognition events in the heterologous *N. benthamiana* expression system, yet this gain-of-function aspect on ID was not sufficient to bring in a new recognition specificity for disease resistance in the endogenous condition. This work is a logical extension of the previous findings on HMA-ID recognition specificity, and the outcome is highly impactful in guiding the research area minded for NLR engineering through ID modifications. The failure to expand the recognition specificity of the NLR-ID RGA5 in rice shall not be perceived as a weak point of the work, but should be considered as a very important piece of solid data that would guide us on future NLR engineering. This work emphasizes how important it is to understand the mode of NLR action in the context of whole molecule as well as of an active NLR signaling complex. With the important first engineering step achieved in this work, namely adapting an ID to embrace a broad physical affinity to a new effector, further engineering is expected to proceed. This direction will surely inform us with intricate controls imposed on NLR activation as well as with evolvability of NLR molecule(s) as a unit in relation to effector recognition.

All the data was clearly presented with compelling graphics and quantitative measurements. Transgenic studies were extensive for the authors to draw a conclusion. The discussion section was in particular well delivering academic merits of this study as well as future perspectives, which I am sure will serve as a milestone in guiding researchers in NLR biology.

Only minor typographical comments can be provided from this reviewer.

Line 142: "has not be" to "has not been"

Line 359: "has" to "have"

Line 369: "hypothesis" to "hypotheses"

Line 400: "has" to "have"

Line 418: "pave" to "paving"

Line 482: empty bracket

Supplemental Table 3: The last six rows seem to lack information on the first five columns.

Reviewer #2 (Remarks to the Author):

The manuscript by Cesari and colleagues reports an important attempt to generate a synthetic NLR that has an extended recognition repertoire of *Magnaporthe oryzae* effectors. Previous structural studies by the author's group and other groups showed that a few effectors of *M. oryzae* interact with the Heavy-Metal-Associated (HMA) domains of two nonhomologous rice NLRs. These two NLRs are RGA5 and Pk1p-1, which are receptors of AVR1-CO39 and AVR-PikD effectors, respectively. RGA5 also detects AVR-Pia in addition to AVR1-CO39. RGA5 and Pk1p-1 function with additional NLRs, RGA4 and Pk2p-2, respectively. While RGA5 and Pk1p-1 appear to specialize in pathogen effector sensing, RGA4 and Pk2p-2 are responsible to execute downstream signaling when their partners NLRs interact with effectors.

As AVR1-CO39 and AVR-PikD interact with different surfaces of the HMA domains of RGA5 and Pk1p-1, the authors generated a synthetic RGA5, of which the HMA domain has both RGA5- and Pk1p-1-type surfaces for the effector binding. In vivo (i.e., co-ip, γ 2h) and in vitro (i.e., surface plasmon resonance) assays provide a strong experimental evidence that the synthetic HMA domain designated HMAm1m2, has a dual binding capacity to the effectors as those of RGA5 and Pk1p-1, meaning that this synthetic domain is able to bind AVR-PikD in addition to AVR1-CO39 (and AVR-Pia).

The subsequent experiment showed that in *N. benthamina* system, the receptor pair of the synthetic RGA5 harboring the modified HMA domain and the wild type RGA4 caused host cell death (an indicative of NLR activation) upon expression of AVR-PikD in addition to expression of AVR1-CO39, suggesting a successful engineering of RGA5. However, while the success in the heterologous system with transient overexpression of effectors and NLRs, in the homologous rice system with presumptive native level expression of NLRs, the stable transgenic rice lines co-expressing the synthetic RGA5 and RGA4 failed to resist *M. oryzae* carrying AVR-PikD despite the retained resistance to *M. oryzae* carrying AVR-CO39 or AVR-Pia. Considering the high K_d of the synthetic HMA domain of RGA5 to the AVR proteins, the interaction of HMA domain with the effectors is insufficient to extend the recognition repertoire of RGA5 in the native condition as the authors discussed.

In conclusion I think that this report is highly valuable to the field of NLR research. Because many NLRs have integrated domains (IDs), which must be involved in recognition of effectors to certain extent, this work alerts us to use of the heterologous overexpression system when examining synthetic NLRs. Thanks to this work, I hope that the use of the homologous system would become a standard in NLR engineering studies.

Below I have listed other concerns on the manuscript.

1) The title and abstract of manuscript

I believe that the interesting and important finding of this manuscript is that the modification of HMA domain alone is insufficient to extend the recognition repertoire of RGA5 leading to immunity. I appreciate very much if the title and abstract explicitly contain this aspect. I understand that such a title may give a negative impression of the work. However, there are already several manuscripts describing “engineering” of NLRs, although they mostly rely on the heterologous system with transient overexpression. In this regard, the current title is not very different from the previous works.

2) Purity and stoichiometry of recombinant proteins

Please provide size exclusion chromatography or equivalent data that have assessed the purity and stoichiometry of recombinant proteins. Do they always exist as monomeric proteins?

3) The rationale of the distinction of HMAM1 and m2

It is not very clear to me why the authors separated HMAM1 and m2 (lines 129-138).

4) Rice cultivar for the transgenic lines

The authors used Nipponbare for the generation of transgenic lines. I’m wondering if this cultivar is suitable to examine the AVR-PikD recognition. For example, if a negative regulator(s) for the AVR-PikD recognition is encoded in the genome of this cultivar, the authors would not be able to detect the AVR-PikD recognition in this cultivar. Have the author’s group or other groups generated a stable transgenic line expressing Pikp-1 and Pikp-2 in Nipponbare and proved that Pikp-1 and Pikp-2 work in this cultivar? If such a transgenic line were unavailable, biolistic delivery or protoplast-mediated transfection of expression constructs to this cultivar would be an option. Such an assay would also help to confirm if a higher expression of receptors (and effectors) could detect AVRs in the homologous system.

5) AVR-Pia binding assay to the HMA domain (SPR)

The association and dissociation patterns of AVR-Pia in Fig 3B are very different from the others. I understand that the mutant version of AVR-Pia exhibited a lower value compared to the wild type. However, this pattern indicates no binding in my view (similar to Figure S4A).

6) Western blot images

The contrast and brightness of images seem to be highly adjusted in some cases. IN general, exaggerated images are inappropriate (e.g., bottom panels in Figure 4 and 5 with Zero background). Please replace images.

7) Line 409

Please describe the MIC clade where Pk1p-1 belongs.

8) Line 482

Something is missing here.

Reviewer #3 (Remarks to the Author):

The manuscript at hand by Cesari et al. attempts at engineering gain-of-function mutation of a plant NLR immune receptor. While the NLR RGA5 from rice has evolved to recognize AVR-CO39 and AVR-Pia effectors of *M. oryzae*, it has naturally no recognition capacity for the effector AVR-PikD. Instead, Pk1p-1 functions as the specific receptor of AVR-PikD. However, Pk1p-1 and RGA5 are both atypical NLRs with an integrated HMA domain and their HMA domains are responsible for effector recognition through direct binding. Most importantly, effector binding of the Pk1p-1 and RGA5 HMA domains is mediated by opposing sides of the HMA surface. The authors took advantage of this circumstance and designed point mutations with the goal to confer AVR-PikD binding capacity to RGA5. The authors created mutations in beta-sheet beta2/beta3 (called m1), beta 4 (m2) and a combinatorial mutant of m1 and m2 (called m1m2). The authors present convincing results from yeast-two-hybrid (Y2H), Surface Plasmon Resonance (SPR) and CoIP from tobacco overexpression to confirm novel binding capacity of RGA5-m1m2 towards AVR-PikD. Moreover, the authors demonstrated nicely that RGA5-m1m2 retains the activity of binding the cognate RGA5 effectors AVR-CO39 and AVR-Pia and thereby demonstrated that AVR-PikD binding of RGA5-m1m2 does not make compromises in the evolved function of RGA5. RGA5 by itself has no inherent direct function to cause cell death, but instead relies on its paired NLR RGA4 to cause cell death. When the authors coexpressed RGA5-m1m2 and RGA4 in tobacco, they observed RGA4-mediated cell death in response to AVR-PikD overexpression as well as in response to AVR-Pia. Up to this point, the results presented are very consistent, statistically sound and thereby very convincing. When the authors moved on with transgenic experiments in rice to employ their engineered RGA5-m1m2 to recognize AVR-PikD, they were unfortunately not able to observe rice immunity towards the Guy1 strain of *M. oryzae*. In the manuscript in its current state, the reasons as to why resistance in rice is not achieved with RGA5-m1m2 remain speculative. My suggestions to improve the manuscript

therefore focus on experimentally explaining the apparent contradiction between the data obtained from transgenic rice and the remaining experiments.

Nevertheless, the authors succeeded in designing an NLR with novel binding capacity and even (indirect) cell-death inducing capacity in plants. The manuscript could serve as a template for future attempts in NLR gain-of-function engineering.

For the above reason and due to the high quality of the presented data I recommend publication.

Still, I would like to ask the authors to consider the following points if they intend to further solidify the evidence:

Major 1: It is very unfortunate that RGA5-m1m2 fails to confer immunity in rice. What I am missing, however, is expression data such as qRT-PCR or if at all feasible Western Blot. Such data could help to explain why RGA5-m1m2 is inactive in rice. Could it be due to low expression level of RGA5-m1m2 compared to RGA5 or even Pikp-1? Without expression data, I am having significant trouble agreeing with the points given in the discussion referring to protein function.

Major 2: Is it at all possible that there are significant differences in the CC domain of RGA4 and Pikp-2 due to coevolution between partners in an NLR pair? Could this maybe explain why the intensity of the resulting immunity is weaker in RGA5-m1m2, despite fully functional binding of RGA5-m1m2 and Avr-Pikp? To answer this question it would be conducive to use a RGA5-PikpHMA control. In addition, further chimeric constructs between Pikp-1 and RGA5 could be made, e.g. a construct including larger parts of the LRR and NOD domain as these could potentially be involved in effector binding.

The following notes address minor points that would certainly improve the quality of the manuscript:

Minor 1 (page 5, line 142): 'has not be...' should be 'has not been...'

Minor 2 (page 6, line 184f): show SDS page results for purified protein

Minor 3 (page 7, 192 ff): How was protein concentration measured? Calculation of dissociation constants is highly dependent on the input amount of protein. Mentioning how concentration was measured and how much was used in absolute numbers would give an indication of the inherent method error.

Minor 4 (page 8, line 246f): 'RGA5m2 recognized AVR-Pia but did not trigger cell death upon co-expression with AVR-PikD'. Please find better wording. In its current phrasing the sentence implies direct action of RGA5 to cause cell death, which we know is only indirect. Also, how do we know it recognizes AVR-Pia, when it does not indirectly cause cell death? Better make a reference to the respective results (CoIP).

Minor 5 (page 11, line 339): 'μ-molar range'. Can here be given an exact number?

Minor 6 (Figure 1, A): Is there maybe a mistake in the positioning of the green box on the very right side (residue 1071)? The motif VEKE of RGA5 and the motif ANKD of Pikp-1 would require a green box under the last residue of the motif rather than the second last (at least according to how the logic for the green box was applied in the rest of the figure).

Minor 7 (Figure 3, A-C): I would suggest to change the naming in the graph legend like this: RGA5_HMA (MBP), RGA5_HMAm1 (MBP), RGA5_HMAm1m2 (MBP)...

Minor 8 (Figure 5, lower panel): the anti-GFP blot of the IP-GFP is quite confusing. It would be helpful for the reader to be given labels on the bands that correspond to the YFP constructs that the authors are trying to detect. Else, at least the molecular weight of the constructs should be given in the figure text.

Minor 9 (figure 6, IP-GFP, anti-HA blot): Why is there a band for the YFP only control in the IP of AVR-Pia:HA and YFP? It would be nice if the authors could briefly discuss this in the results.

Minor 10 (Figure 7, A): There is a slight increase of cell death when comparing RGA5/RGA4 and RGA4/RGA5m1m2 (left and right image at the top). Could the authors please discuss if they agree with this observation? Is it possible that RGA5m1m2 is slightly less efficient at suppressing RGA4-dependent cell death in the absence of the effector? Can the same observation also be made in B? Although it is apparently not statistically significant.

Spelling and wording are generally on a high level with one exception already mentioned in 'minor 1'.

REVIEWER COMMENTS

Reviewer #1 (Remarks to the Author):

The manuscript authored by Cesari and colleagues presents the extensive body of work describing their attempts to expand NLR recognition specificity through a structure-guided engineering of integrated decoy domain (ID). Using the well-studied HMA ID and its cognate binding AVR partners, the authors demonstrated that new binding affinity to a specific effector could be achieved through structure-guided substitutions. The expansion of physical affinity was correlated with recognition events in the heterologous *N. benthamiana* expression system, yet this gain-of-function aspect on ID was not sufficient to bring in a new recognition specificity for disease resistance in the endogenous condition. This work is a logical extension of the previous findings on HMA-ID recognition specificity, and the outcome is highly impactful in guiding the research area minded for NLR engineering through ID modifications. The failure to expand the recognition specificity of the NLR-ID RGA5 in rice shall not be perceived as a weak point of the work, but should be considered as a very important piece of solid data that would guide us on future NLR engineering. This work emphasizes how important it is to understand the mode of NLR action in the context of whole molecule as well as of an active NLR signaling complex. With the important first engineering step achieved in this work, namely adapting an ID to embrace a broad physical affinity to a new effector, further engineering is expected to proceed. This direction will surely inform us with intricate controls imposed on NLR activation as well as with evolvability of NLR molecule(s) as a unit in relation to effector recognition.

All the data was clearly presented with compelling graphics and quantitative measurements. Transgenic studies were extensive for the authors to draw a conclusion. The discussion section was in particular well delivering academic merits of this study as well as future perspectives, which I am sure will serve as a milestone in guiding researchers in NLR biology.

Only minor typographical comments can be provided from this reviewer.

Line 142: “has not be” to “has not been”

Line 359: “has” to “have”

Line 369: “hypothesis” to “hypotheses”

Line 400: “has” to “have”

Line 418: “pave” to “paving”

Line 482: empty bracket

Response: All typos have been corrected and brackets have been removed.

Supplemental Table 3: The last six rows seem to lack information on the first five columns.

Response: This error has been corrected and a column has been added for the “rice cultivar”. The error came from the transformation of the original Excel file into a pdf document.

Reviewer #2 (Remarks to the Author):

The manuscript by Cesari and colleagues reports an important attempt to generate a synthetic NLR that has an extended recognition repertoire of *Magnaporthe oryzae* effectors. Previous structural studies by the author’s group and other groups showed that a few effectors of *M. oryzae* interact with the Heavy-Metal-Associated (HMA) domains of two nonhomologous rice NLRs. These two NLRs are RGA5 and Pikip-1, which are receptors of AVR1-CO39 and AVR-

PikD effectors, respectively. RGA5 also detects AVR-Pia in addition to AVR1-CO39. RGA5 and Pikp-1 function with additional NLRs, RGA4 and Pikp-2, respectively. While RGA5 and Pikp-1 appear to specialize in pathogen effector sensing, RGA4 and Pikp-2 are responsible to execute downstream signaling when their partners NLRs interact with effectors.

As AVR1-CO39 and AVR-PikD interact with different surfaces of the HMA domains of RGA5 and Pikp-1, the authors generated a synthetic RGA5, of which the HMA domain has both RGA5- and Pikp-1-type surfaces for the effector binding. In vivo (i.e., co-ip, y2h) and in vitro (i.e., surface plasmon resonance) assays provide a strong experimental evidence that the synthetic HMA domain designated HMAm1m2, has a dual binding capacity to the effectors as those of RGA5 and Pikp-1, meaning that this synthetic domain is able to bind AVR-PikD in addition to AVR1-CO39 (and AVR-Pia).

The subsequent experiment showed that in *N. benthamina* system, the receptor pair of the synthetic RGA5 harboring the modified HMA domain and the wild type RGA4 caused host cell death (an indicative of NLR activation) upon expression of AVR-PikD in addition to expression of AVR1-CO39, suggesting a successful engineering of RGA5. However, while the success in the heterologous system with transient overexpression of effectors and NLRs, in the homologous rice system with presumptive native level expression of NLRs, the stable transgenic rice lines co-expressing the synthetic RGA5 and RGA4 failed to resist *M. oryzae* carrying AVR-PikD despite the retained resistance to *M. oryzae* carrying AVR-CO39 or AVR-Pia. Considering the high K_d of the synthetic HMA domain of RGA5 to the AVR proteins, the interaction of HMA domain with the effectors is insufficient to extend the recognition repertoire of RGA5 in the native condition as the authors discussed.

In conclusion I think that this report is highly valuable to the field of NLR research. Because many NLRs have integrated domains (IDs), which must be involved in recognition of effectors to certain extent, this work alerts us to use of the heterologous overexpression system when examining synthetic NLRs. Thanks to this work, I hope that the use of the homologous system would become a standard in NLR engineering studies.

Below I have listed other concerns on the manuscript.

1) The title and abstract of manuscript

I believe that the interesting and important finding of this manuscript is that the modification of HMA domain alone is insufficient to extend the recognition repertoire of RGA5 leading to immunity. I appreciate very much if the title and abstract explicitly contain this aspect. I understand that such a title may give a negative impression of the work. However, there are already several manuscripts describing “engineering” of NLRs, although they mostly rely on the heterologous system with transient overexpression. In this regard, the current title is not very different from the previous works.

Response: We agree with this comment and have made according changes to the title, which is now: *“Molecular engineering of the integrated decoy domain in a plant NLR immune receptor creates a new effector recognition specificity but does not confer immunity.”*

We have also clarified this in the last sentences of the abstract:

Therefore, our study provides a proof of concept for the design of new effector recognition specificities in NLRs through molecular engineering of IDs. However, these modifications were insufficient to create a novel blast resistance specificity in transgenic rice plants. This pinpoints

significant knowledge gaps that limit the full deployment of this NLR-ID engineering strategy and provides hypotheses for future research on this topic.

2) Purity and stoichiometry of recombinant proteins

Please provide size exclusion chromatography or equivalent data that have assessed the purity and stoichiometry of recombinant proteins. Do they always exist as monomeric proteins?

Response: SDS page analysis for all recombinant proteins and size exclusion chromatography for the MBP fusions of RGA5_HMA, RGA5_HMAm1m2 and Pk1p_HMA have been added as Supplemental Figure 4. The recombinant wild-type and mutant AVR effectors (identical constructs) have already previously been shown to be monomeric (de Guillen et al., 2015, Guo et al., 2018).

3) The rationale of the distinction of HMAm1 and m2

It is not very clear to me why the authors separated HMAm1 and m2 (lines 129-138).

Response: We generated individual HMAm1 and HMAm2 mutants in addition to the HMAm1m2 mutant to minimize changes in the RGA5_HMA domain in our attempt to achieve efficient AVR-PikD binding. We thought this would reduce the risk of eventual deleterious effects due to changes in the RGA5_HMA sequence. In addition, we also wanted to test the contribution of the individual binding surfaces to high affinity AVR-PikD binding.

4) Rice cultivar for the transgenic lines

The authors used Nipponbare for the generation of transgenic lines. I'm wondering if this cultivar is suitable to examine the AVR-PikD recognition. For example, if a negative regulator(s) for the AVR-PikD recognition is encoded in the genome of this cultivar, the authors would not be able to detect the AVR-PikD recognition in this cultivar. Have the author's group or other groups generated a stable transgenic line expressing Pk1p-1 and Pk1p-2 in Nipponbare and proved that Pk1p-1 and Pk1p-2 work in this cultivar? If such a transgenic line were unavailable, biolistic delivery or protoplast-mediated transfection of expression constructs to this cultivar would be an option. Such an assay would also help to confirm if a higher expression of receptors (and effectors) could detect AVRs in the homologous system.

Response: Stable transgenic Nipponbare rice lines expressing Pk1p-1 and Pk1p-2 have not been described in the literature. However, the allelic and highly homologous genes *Pik-1/Pik-2* and *Pikm-1/Pikm-2* were shown to be functional for AVR-Pik recognition in transgenic Nipponbare lines (Ashikawa et al., 2010 DOI: 10.1007/s00122-010-1506-3, Ashikawa et al., 2008 DOI: 10.1534/genetics.108.095034). Both allelic pairs, *Pik* and *Pikm*, recognize AVR-PikD as well as additional AVR-Pik alleles as they possess extended AVR-Pik recognition specificities as compared to *Pk1p* (Kanzaki et al. 2012, DOI: 10.1111/j.1365-313X.2012.05110.x). This excludes that there are negative regulators for AVR-PikD recognition encoded in the Nipponbare genome.

5) AVR-Pia binding assay to the HMA domain (SPR)

The association and dissociation patterns of AVR-Pia in Fig 3B are very different from the others. I understand that the mutant version of AVR-Pia exhibited a lower value compared to the wild type. However, this pattern indicates no binding in my view (similar to Figure S4A).

Response: Indeed the SPR curves observed for the binding of AVR-Pia to the different HMA domains are different from the typical sensorgrams reported for molecular interactions

monitored by SPR, usually in the nano-molar range. Nonetheless, the AVR-Pia/HMAs interactions observed by SPR are specific and the “mesa shape” binding profiles are in fact typical of those expected for μ -molar interactions between an analyte and an immobilized ligand (see the file named “Figures for reviewing only”, which contains 4 figures).

Figure R1 shows the results of a kinetic titration experiment (not reported in the manuscript) that we have performed by injecting increasing concentration of purified wild-type AVR-Pia on different HMAs fused to MBP and captured by anti-MBP antibodies immobilized on the sensor-chip (similar to the titration experiment we performed with AVR-PikD and reported in supplemental Figure 5). The binding response (responsive units or RU) recorded at the highest injected concentrations of effector (10 μ M) is over 20 times higher for the MBP:HMA fusions (~400-500 RU) than that observed for the negative control (MBP alone, ~15 RU). In the single shot experiments that we performed for comparing the binding of AVR effectors injected at 1 μ M (Figure 3B of the manuscript and Figure R2A of the present response), we used an inactive effector variant (AVR-Pia_F24S) as a negative control and observed a 20 to 40 fold reduction in RU compared to that observed with wild-type AVR-Pia. This demonstrates that the SPR signal observed with WT AVR-Pia results from specific binding to HMA domains, rather than from an artifact that could be due to a bulk effect (in spite of the multi-channel set-up used for automatic subtraction of a reference channel).

We also obtained very similar results when comparing the normalized binding response calculated for two independent experiments, performed under similar conditions but using different protein preparations of wild-type AVR-Pia (Figure R2).

Although accurate kinetic parameters (k_a and k_d) cannot be retrieved in case of low affinity binding reactions, it is possible to perform data analysis at steady state (usually reached within a few seconds after injection for low affinity binding). Evaluation of dissociation constants ($K_D=k_a/k_d$) is done by plotting the binding response measured at increasing concentrations, as shown in Figure R4. The K_D value estimated for AVR-Pia/RGA5_HMA by this method (7.4 μ M) is matching surprisingly well the K_D value we previously measured for this complex by ITC (7.8 μ M, Ortiz et al., 2017).). Similarly, the affinity constant of 4.9 μ M we estimated for the binding of AVR1-CO39 to RGA5_HMA from the steady state analysis of SPR data is also in very good accordance with that measured previously by ITC (7.2 μ M, Guo et al. 2018), as stated now more explicitly in the discussion of the main text.

Finally, as seen in Figure R4, the simulated SPR sensorgram based on the affinity constant estimated at steady state for AVR-PikD/ RGA5_HMA interaction reproduces very well the experimental binding curve.

We are therefore confident that the “mesa shape” profiles we observed for some AVR/HMA interactions indicate weak, yet real and specific, binding. To make that point clearer to the reader, we have modified the sentence reporting these results in the result section: “Weak yet specific binding was observed for the interaction of either AVR1-CO39 or AVR-Pia to RGA5_HMA mutants and wild type (Figure 3A, B and D), consistent with the affinity constants in the μ molar range previously reported for the binding of RGA5_HMA for both ligands.”

6) Western blot images

The contrast and brightness of images seem to be highly adjusted in some cases. In general, exaggerated images are inappropriate (e.g., bottom panels in Figure 4 and 5 with Zero background). Please replace images.

Response: The contrast and brightness of western blot images have not been adjusted. We provide, enclosed to this resubmission, the raw images of the blots (including images of signals detected by chemiluminescence, pictures of the corresponding membranes after detection and Ponceau staining of the membranes) where this can be verified.

7) Line 409

Please describe the MIC clade where Pikp-1 belongs.

Response: The NLR clade to which Pikp-1 belongs is not enriched for integrated domains and Pikp-1 is in this sense an exception in its clade. Only 3 NLR clades enriched for integrated domains have been identified in cereal genomes and were called MICs in Bailey et al., 2016.

We therefore have added the sentence: "*Pikp-1 belongs to another NLR clade that shows much less frequent ID integration.*"

8) Line 482

Something is missing here.

Response: Empty brackets have been removed.

Reviewer #3 (Remarks to the Author):

The manuscript at hand by Cesari et al. attempts at engineering gain-of-function mutation of a plant NLR immune receptor. While the NLR RGA5 from rice has evolved to recognize AVR-CO39 and AVR-Pia effectors of *M. oryzae*, it has naturally no recognition capacity for the effector AVR-PikD. Instead, Pikp-1 functions as the specific receptor of AVR-PikD. However, Pikp-1 and RGA5 are both atypical NLRs with an integrated HMA domain and their HMA domains are responsible for effector recognition through direct binding. Most importantly, effector binding of the Pikp-1 and RGA5 HMA domains is mediated by opposing sides of the HMA surface. The authors took advantage of this circumstance and designed point mutations with the goal to confer AVR-PikD binding capacity to RGA5. The authors created mutations in beta-sheet beta2/beta3 (called m1), beta 4 (m2) and a combinatorial mutant of m1 and m2 (called m1m2). The authors present convincing results from yeast-two-hybrid (Y2H), Surface Plasmon Resonance (SPR) and CoIP from tobacco overexpression to confirm novel binding capacity of RGA5-m1m2 towards AVR-PikD. Moreover, the authors demonstrated nicely that RGA5-m1m2 retains the activity of binding the cognate RGA5 effectors AVR-CO39 and AVR-Pia and thereby demonstrated that AVR-PikD binding of RGA5-m1m2 does not make compromises in the evolved function of RGA5. RGA5 by itself has no inherent direct function to cause cell death, but instead relies on its paired NLR RGA4 to cause cell death. When the authors coexpressed RGA5-m1m2 and RGA4 in tobacco, they observed RGA4-mediated cell death in response to AVR-PikD overexpression as well as in response to AVR-Pia. Up to this point, the results presented are very consistent, statistically sound and thereby very convincing. When the authors moved on with transgenic experiments in rice to employ their engineered RGA5-m1m2 to recognize AVR-PikD, they were unfortunately not able to observe rice immunity towards the Guy1 strain of *M. oryzae*.

In the manuscript in its current state, the reasons as to why resistance in rice is not achieved with RGA5-m1m2 remain speculative. My suggestions to improve the manuscript therefore focus on experimentally explaining the apparent contradiction between the data obtained from transgenic rice and the remaining experiments.

Nevertheless, the authors succeeded in designing an NLR with novel binding capacity and even (indirect) cell-death inducing capacity in plants. The manuscript could serve as a template for future attempts in NLR gain-of-function engineering.

For the above reason and due to the high quality of the presented data I recommend publication. Still, I would like to ask the authors to consider the following points if they intend to further solidify the evidence:

Major 1: It is very unfortunate that RGA5-m1m2 fails to confer immunity in rice. What I am missing, however, is expression data such as qRT-PCR or if at all feasible Western Blot. Such data could help to explain why RGA5-m1m2 is inactive in rice. Could it be due to low expression level of RGA5-m1m2 compared to RGA5 or even Pkp-1? Without expression data, I am having significant trouble agreeing with the points given in the discussion referring to protein function.

Response: We performed analysis of transgene expression (*RGA5* wild-type and mutants as well as *RGA4*) by qRT-PCR. This data has been added as Supplemental Figure 11. Overall, we observe that the relative expressions of *RGA4* and *RGA5* are quite variable between the different RGA4+RGA5 (wild-type or mutant) transgenic lines. For instance, the RGA4+RGA5_L12 line shows rather weak expression of both genes while the RGA4+RGA5_L2 line expresses *RGA5* at high level. Both these transgenic lines are fully resistant to Guy11-AVR-*Pia* indicating that even the weak expression of *RGA4* and *RGA5* is sufficient for AVR-*Pia* recognition and induction of immunity. In the RGA4+RGA5 mutant lines, the expression of *RGA4* and *RGA5* are either similar to that of the RGA4+RGA5_L12 line or higher (at similar level as in the rice variety Kitaake that carries *RGA4* and *RGA5*). This indicates that these RGA4+RGA5 mutant lines properly express *RGA4* and *RGA5*.

In addition, data of infection assays presented in the initial submission show that transgenic rice lines carrying *RGA5* mutant constructs have the same level of resistance to Guy11-AVR-*Pia* and Guy11-AVR1-*CO39* as lines carrying *RGA5* wild-type.

Taken together, these data support the conclusion that lack of resistance to Guy11-AVR-*PikD* in transgenic *RGA5* mutant lines is not due to low transgene expression but to failure of efficient effector recognition in the transgenic rice plants.

Major 2: Is it at all possible that there are significant differences in the CC domain of RGA4 and Pkp-2 due to coevolution between partners in an NLR pair? Could this maybe explain why the intensity of the resulting immunity is weaker in RGA5-m1m2, despite fully functional binding of RGA5-m1m2 and Avr-Pkp? To answer this question it would be conducive to use a RGA5-PkpHMA control. In addition, further chimeric constructs between Pkp-1 and RGA5 could be made, e.g. a construct including larger parts of the LRR and NOD domain as these could potentially be involved in effector binding.

Response: RGA4 and Pkp-2 are from different NLR clades and show indeed sequence and functional differences (e.g. RGA4 is auto active when overexpressed alone while Pkp-2 is not). However, the immune response triggered by RGA4/RGA5 is at least as strong as the one activated by Pkp-1/Pkp-2 as, in both cases, complete blast resistance characterized by lack of visible symptoms or by small HR lesions is observed.

We did not generate the RGA5-PkpHMA construct since there is a risk that integration of the entire Pkp1-HMA domain hampers proper function of RGA5 and results in auto-activity upon co-expression with RGA4. Besides, our idea was to build on the knowledge provided by the previous structural analyses and design precise structure-informed replacements in the HMA domain of RGA5 rather than entire domain swaps. Since our RGA5-HMA-m1m2 construct binds AVR-PikD with high affinity, we do not see the additional insight that the RGA5-PkpHMA construct would provide.

Present data suggest that AVR-PikD binds exclusively to the HMA domain of Pikp-1. To our knowledge, there is no indication for the interaction of this effector with additional domains of Pikp-1. Should such additional interactions exist, there is a high probability that their transfer, or transfer of extended parts of Pikp-1 into RGA5, would not lead to better effector recognition and would rather interfere with proper interaction with RGA4. Indeed, there is good evidence that the mechanisms of effector recognition and sensor/helper NLR interactions differ in RGA4/RGA5 and Pikp-1/Pikp-2 (e.g. Pikp-1 and RGA5 are from different NLR clades and have different sites of HMA integration; see above comment for RGA4 and Pikp-2). Our aim in this study was to test if replacements in the integrated domain alone would be sufficient to generate a new recognition specificity, not to change completely the mode of action of RGA5. However, we agree with Reviewer 3 that it will be interesting to elucidate in future studies why the RGA5_HMA_{m1m2} mutant does not efficiently recognize AVR-PikD in transgenic rice plants. This involves the investigation of the role of the other RGA5 domains and their action and interaction in the full-length protein. Such analyses go beyond the scope of this study since it requires extensive analysis of important number of deletion and mutant constructs as well as the generation of new transgenic rice lines (which takes ~1 year, including characterization).

The following notes address minor points that would certainly improve the quality of the manuscript:

Minor 1 (page 5, line 142): ‘has not be...’ should be ‘has not been...’

Response: The sentence has been corrected.

Minor 2 (page 6, line 184f): show SDS page results for purified protein

Response: Now shown in Supplemental Figure 4.

Minor 3 (page 7, 192 ff): How was protein concentration measured? Calculation of dissociation constants is highly dependent on the input amount of protein. Mentioning how concentration was measured and how much was used in absolute numbers would give an indication of the inherent method error.

Response: This information was now added to the relevant Methods section: «Protein concentration was determined by absorption of UV light at 280 nm using a Nano Drop and theoretical extinction coefficients.»

Minor 4 (page 8, line 246f): ‘RGA5_{m2} recognized AVR-Pia but did not trigger cell death upon co-expression with AVR-PikD’. Please find better wording. In its current phrasing the sentence implies direct action of RGA5 to cause cell death, which we know is only indirect. Also, how do we know it recognizes AVR-Pia, when it does not indirectly cause cell death? Better make a reference to the respective results (CoIP).

Response: We have changed the sentence to “RGA5_{m2} recognized AVR-Pia but not AVR-PikD.”

Minor 5 (page 11, line 339): ‘μ-molar range’. Can here be given an exact number?

Response: We replaced this and indicated the K_D values estimated from SPR data analysis at steady state.

Minor 6 (Figure 1, A): Is there maybe a mistake in the positioning of the green box on the very right side (residue 1071)? The motif VEKE of RGA5 and the motif ANKD of Pikp-1 would require a green box under the last residue of the motif rather than the second last (at least according to how the logic for the green box was applied in the rest of the figure).

Response: There was indeed a mistake. It has been corrected in Figure 1 and Supplemental Figure 1.

Minor 7 (Figure 3, A-C): I would suggest to change the naming in the graph legend like this: RGA5_HMA (MBP), RGA5_HMAm1 (MBP), RGA5_HMAm1m2 (MBP)...

Response: We have changed the naming in the graph legends as suggested by Reviewer 3.

Minor 8 (Figure 5, lower panel): the anti-GFP blot of the IP-GFP is quite confusing. It would be helpful for the reader to be given labels on the bands that correspond to the YFP constructs that the authors are trying to detect. Else, at least the molecular weight of the constructs should be given in the figure text.

Response: Labels (asterisks) on the bands have been added to the anti-GFP blot of the IP-GFP. The following sentence has also been added to the figure legend: *“In the IP-GFP/α-GFP panel, yellow and white asterisks indicate the YFP:Pikp-1_HMA and YFP proteins respectively, while blue asterisks show the YFP:RGA5 wild type and mutant proteins.”*

Minor 9 (figure 6, IP-GFP, anti-HA blot): Why is there a band for the YFP only control in the IP of AVR-Pia:HA and YFP? It would be nice if the authors could briefly discuss this in the results.

Response: There is indeed a very weak unspecific interaction between AVR-Pia:HA and YFP that is also visible in Figure 5. In addition, there may be a bit of spillover from the adjacent line with very strong AVR-Pia signal.

Minor 10 (Figure 7, A): There is a slight increase of cell death when comparing RGA5/RGA4 and RGA4/RGA5m1m2 (left and right image at the top). Could the authors please discuss if they agree with this observation? Is it possible that RGA5m1m2 is slightly less efficient at suppressing RGA4-dependent cell death in the absence of the effector? Can the same observation also be made in B? Although it is apparently not statistically significant.

Response: Quantitative analyses with a large number of individual measurements and different biological replicate experiments shown in Figure 7B and Supplemental Figure 7 do not support a difference in cell death suppressor activity between RGA5m1m2 and RGA5 wild type. Therefore, the slight differences in the images in panel A should not be over interpreted and rather reflect the variability of the assay.

Spelling and wording are generally on a high level with one exception already mentioned in ‘minor 1’.

REVIEWERS' COMMENTS

Reviewer #1 (Remarks to the Author):

This reviewer is satisfied with the revised manuscript, which addressed all the reviewers' comments in a rigorous manner. As pointed out in the previous comment, the lack of immunity by the engineered ID-RGA5 with the expanded choice of effector levels up our understanding in receptor function. Thus, I do appreciate that the revised manuscript put forward this notion in the title.

minor point

L474 Page 14;

I rather think "communalities" would mean "commonalities" as compared to "specificity".

Reviewer #2 (Remarks to the Author):

The authors have addressed the points raised in my previous review.

Reviewer #3 (Remarks to the Author):

The authors have addressed all of my comments and the manuscript has improved significantly by the additions. Therefore, I recommend publication of the manuscript.

We thank the reviewers for their constructive comments throughout the entire reviewing process that helped us to improve significantly the manuscript.

REVIEWERS' COMMENTS

Reviewer #1 (Remarks to the Author):

This reviewer is satisfied with the revised manuscript, which addressed all the reviewers' comments in a rigorous manner. As pointed out in the previous comment, the lack of immunity by the engineered ID-RGA5 with the expanded choice of effector levels up our understanding in receptor function. Thus, I do appreciate that the revised manuscript put forward this notion in the title.

minor point

L474 Page 14;

I rather think "communalities" would mean "commonalities" as compared to "specificity".

"communalities" has been replaced by "commonalities"

Reviewer #2 (Remarks to the Author):

The authors have addressed the points raised in my previous review.

Reviewer #3 (Remarks to the Author):

The authors have addressed all of my comments and the manuscript has improved significantly by the additions. Therefore, I recommend publication of the manuscript.